# REAL: REtrieval-Augmented and Logic-constructed Attention Behaviors for Robust KV Cache Compression

## Abstract

The growing input sequence length of large language models (LLMs) places increasing pressure on key-value (KV) cache storage, making efficient inference challenging. Existing retrieval-based compression methods neglect the impact of distracted, biased, and widespread attention behaviors, raising *robustness* concerns. To address these challenges, this paper proposes **REtrieval-Augmented and Logic-constructed (REAL)** KV cache compression that implements a robust, low-cost, training-free method, capturing diverse attention behaviors. REAL introduces an *attention weight confusion matrix (AWCM)* to categorize attention behaviors and an *inference score (INFsc)* that balances retrieval and logic for head-wise dynamic budget allocation with an empirical per-layer safeguard. Experiments on long-sequence QA and non-QA tasks show that REAL achieves more robust compression than state-of-the-art baselines and even surpasses FullKV in certain situations. To our knowledge, REAL is the first approach to compress KV caches by attention behavior analysis, offering a new perspective.

## 1 Introduction

Retrieval-driven and logic-faithful Transformer-based (Vaswani et al., 2017) large language models (LLMs) (OpenAI, 2025; Anthropic, 2025; Huang & Yang, 2025) have shown remarkable performance in tasks such as question answering (QA) (Kamalloo et al., 2023; Jiang et al., 2021; Su et al., 2019) and code generation (Rozière et al., 2023). To speed up inference, the models rely on *key-value (KV) caches*, which store key and value vectors to avoid recalculations. However, as the size of a KV cache grows with sequence length, model dimensionality, and batch size, it quickly overwhelms the memory capacity of graphics processing units (GPUs). For example, the cache size for 64 x 4,096 tokens in GPT-3 (Brown et al., 2020) requires about 1,208GB, whereas an NVIDIA H200 GPU has only 141GB of device memory. Given these constraints, efficient KV cache compression has become essential.

Regarding sequence length and model dimensionality, related literature has primarily focused on *quantization* and *retrieval*. Quantization methods include KVTuner (Li et al., 2025b), Cache Me If You Must (Shutova et al., 2025), CommVQ (Li et al., 2025a), and QuantSpec (Tiwari et al.,

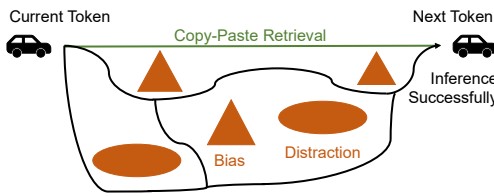

Figure 1: Conceptual illustration showing that attention does not always follow a direct retrieval path between tokens and must analyze both passive distraction and active bias.

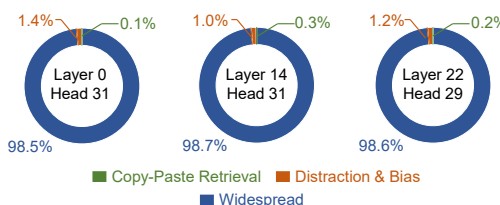

Figure 2: A single head in the Llama-3-8B-Instruct model (Grattafiori et al., 2024) exhibits diverse attention behaviors, as measured by NIAH token counting (Kamradt, 2023).

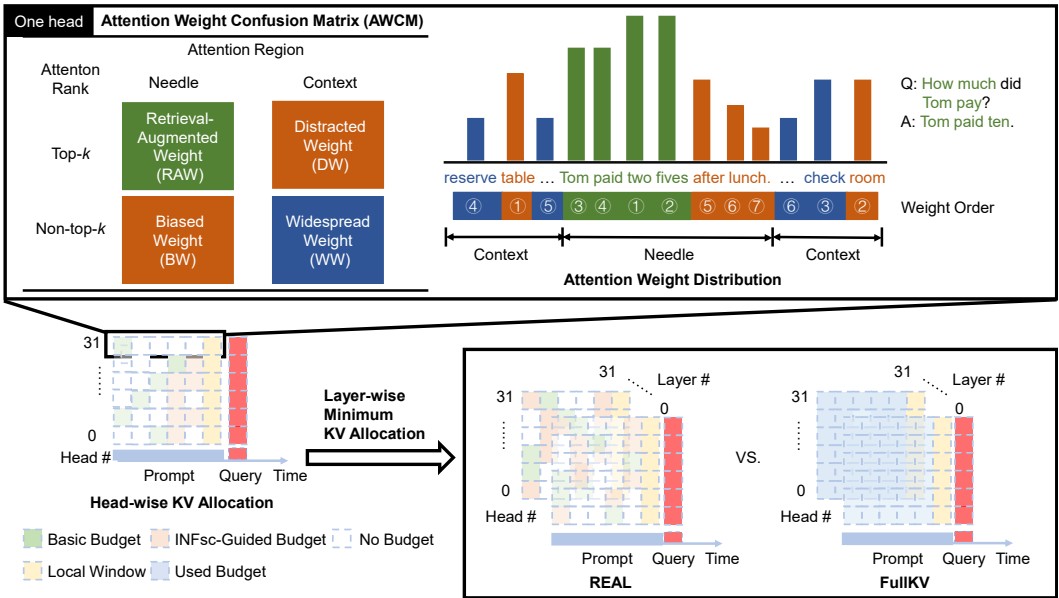

Figure 3: The REAL method operates in two stages: i) Head-wise allocation (upper part): The attention weight confusion matrix (AWCM) guides head-wise KV allocation based on each head's basic budget with intentionally designed needles. For example, "two fives" (2 x 5) is correctly inferred as a payment (10), despite "after lunch" inducing an active time-related bias (e.g., 2:05) and "room" or "table" introducing passive location-related distractions (e.g., 25). Within each head, KV entries are selected to preserve the local window and its most relevant prior tokens. ii) Layer-aware allocation (lower-right part): A minimal empirical KV budget is assigned to the head with the lowest inference score (INFsc) in each layer, serving as a stacked-layer safeguard to preserve robustness under KV compression.

2025). Retrieval methods, generally based on copy-paste mechanisms, can be divided into token-wise position-driven (Zhang et al., 2023; Shi et al., 2025; Nguyen et al., 2025; Liu et al., 2023; Adnan et al., 2024; Ghadia et al., 2025; Xiao et al., 2024; Zhang et al., 2023), layer-wise distribution-driven (Cai et al., 2024; Wang et al., 2025; Tang et al., 2025b; Zhong et al., 2025; Qin et al., 2025; Liu et al., 2024; Nguyen et al., 2025), head-wise Needle-in-a-Haystack (NIAH)-driven (Kamradt, 2023; Wu et al., 2025; Fu et al., 2025; Xiao et al., 2025), and head-wise pattern-driven (Li et al., 2024b; Tang et al., 2025a; Zhang et al., 2025) approaches. These methods help GPUs support larger batch sizes and thereby improve overall performance.

During continuous generation, prior methods have two critical problems. *First, multiple attention behaviors across the head dimension make a single behavior per head fragile.* Long sequences require a resilient inference path. While head-wise NIAH-driven methods (Wu et al., 2025; Fu et al., 2025; Xiao et al., 2025) are more fine-grained and outperform token- and layer-wise approaches, retrieval heads, as shown in Figure 1, often degenerate into simple copy-paste operations from the needle in semantically irrelevant contexts (Wu et al., 2025), neglecting successful inference by analyzing active bias and passive contextual distraction behaviors. This limitation exposes a potential threat to the *robustness* of KV cache compression. To verify the presence of diverse attention behaviors, we constructed needles representative of the upper part in Figure 3 and inserted them at 33 depths (ranging from 2% to 98% in 3% increments) across 30 context lengths (from 1,024 to 30,720 tokens in steps of 1,024). To account for fluctuations in key token rankings (e.g., when a token shifts from 1st to 2nd), we averaged token counts for both top-$k$ (i.e., retrieval-augmented) and non-top-$k$ (i.e., biased) attention across insertion depths and context lengths, capturing distracted and widespread behaviors. Conditioned on successful inference, this evaluation confirms the diversity of attention behaviors across heads, as plotted in Figure 2.

*Second, Stacked-layers should be safeguarded to preserve robustness.* Layer-wise methods typically divide models into shallow (e.g., Llama-3-8B-Instruct (Grattafiori et al., 2024) layers 0–9), middle (10–19), and deep (20–31) parts (Cai et al., 2024; Tang et al., 2025b; Wang et al., 2025). This



Figure 4: Heatmaps of LCsc, RAsc, and INFsc on Llama-3-8B-Instruct. The three extremely low scores in Layer 1 disclose the limitations of the oversimplified early-middle-deep layer division used in previous work.

rigid partitioning oversimplifies the model, overlooking the unique role of each layer. For example, both Layers 0 and 1 attend globally by position yet diverge in focus: Layer 0 emphasizes topic-relevant content (e.g., "This novel tells a story about growth"), whereas Layer 1 highlights off-topic details (e.g., "One day, the sun shines on the mountain"). Head-wise methods, originating from a layer-centric perspective, only address robustness across parallel heads and fail to ensure robust information transfer across serially attacked layers (Vaswani et al., 2017). To resolve these challenges, our approach compresses the KV cache robustly across both sequential length and model dimensionality during the prefill phase.

This paper introduces **REtrieval-Augmented and Logic-constructed (REAL) KV cache compression**, illustrated in Figure 3. REAL is a robust, low-cost, training-free method compatible with FlashAttention (Dao, 2024) and GQA (Ainslie et al., 2023) that leverages attention behaviors with minimal model modifications, without compromising accuracy as in FullKV. To the best of our knowledge, REAL is the first algorithm to achieve robust KV cache compression by incorporating attention behavior analysis through a confusion matrix (Susmaga, 2004). The following outlines the design and key contributions of our method.

The REAL KV cache compression accounts for *comprehensive attention behaviors across head dimension*. Inspired by Signal-to-Noise Ratio in information theory (Shannon, 1948), the fine-grained decomposition of confusion matrices (Susmaga, 2004), and the dual-process theory of fast and slow thinking (Kahneman, 2011), we conceptualize attention behaviors as a comparative process. Copy-paste retrieval reflects fast localization, whereas active bias and passive distraction degrade inference performance. Accordingly, we utilize comparative normalized attention weights instead of raw attention scores. As a result, we build the **attention weight confusion matrix (AWCM)**, as illustrated in the upper left part of Figure 3, with weights for *retrieval-augmented*, *biased*, *distracted*, and *widespread* to provide a general diagnostic. Similar to precision and recall(Susmaga, 2004) we define two metrics: the **retrieval-augmented score (RAsc)** and **logic-constructed score (LCsc)**. Both metrics consider retrieval-augmented behavior that directly supports inference. In addition, RAsc measures distraction, while LCsc analyzes bias. As shown in Figure 4, these scores vary significantly across heads and layers, underscoring the need for balanced evaluation. We therefore propose the **inference score (INFsc)** (Section 2.3), defined as the harmonic mean of RAsc and LCsc. Within each head's kv allocation, we adopt a per-head KV entry selection strategy that preserves the local attention window while selecting the most relevant prior tokens, following SnapKV (Li et al., 2024b) (Section 2.5). Long sequences require a resilient inference path. Along the layer dimension, we allocate a small empirical base budget of $\gamma$ to each layer to preserve its unique role and maintain continuity across stacked layers.

Evaluations across diverse benchmarks show that REAL outperforms state-of-the-art methods under the identical KV size constraint. Downstream tasks require capabilities in degrading distraction and bias analysis, where REAL demonstrates clear superiority. Moreover, REAL reduces both memory footprint and inference latency compared with FullKV in prefill stage. Intuitively, logic construction may further enhance retrieval as manifested in the relationship between LCsc and RAsc in Figure 5, but this is orthogonal to the main scope of this paper and thus not explored in detail.

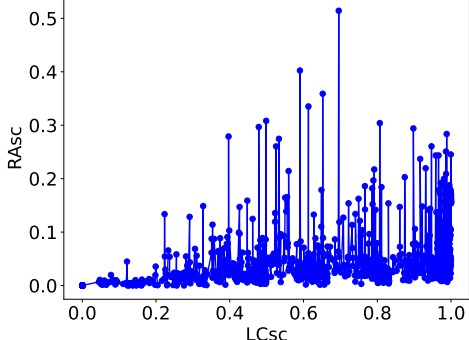

Figure 5: RAsc–LCsc Relationship.

## 2 ADAPTIVE KV BUDGET ALLOCATION METHOD

In this section, we describe i) construction of needles, ii) definition of four attention behaviors, iii) the computation of the RAsc, LCsc, and INFsc metrics, iv) KV budget allocation to individual attention heads based on INFsc, while assigning each layer an empirical base budget of $\gamma$ to ensure stable information flow across stacked layers.

### 2.1 CONSTRUCTION OF NEEDLES

In real-world long documents, web pages, or complex dialogues, useful information is surrounded by a large amount of background and irrelevant content. To simulate information reasoning over long contexts (e.g., a Paul Graham essay), we manually construct novel needle cases that the model has never encountered, as shown in Figure 6. The design ensures the model must rely on the KV cache to get knowledge from the input sequence, rather than falling back on internally knowledge learned during pre-training. Referencing the experimental setup (Wu et al., 2025), we uniformly sampled 30 different sequence lengths within the haystack, ranging from 1K to 30K tokens. For each sequence length, the query was inserted into the haystack, which is irrelevant to the needle, at 33 uniform positions tested between 2% and 98% of the sequence length with a 3% step length, covering various positions from beginning to end. This design allows us to evaluate the model's inference capability across different context depths and diverse contexts.

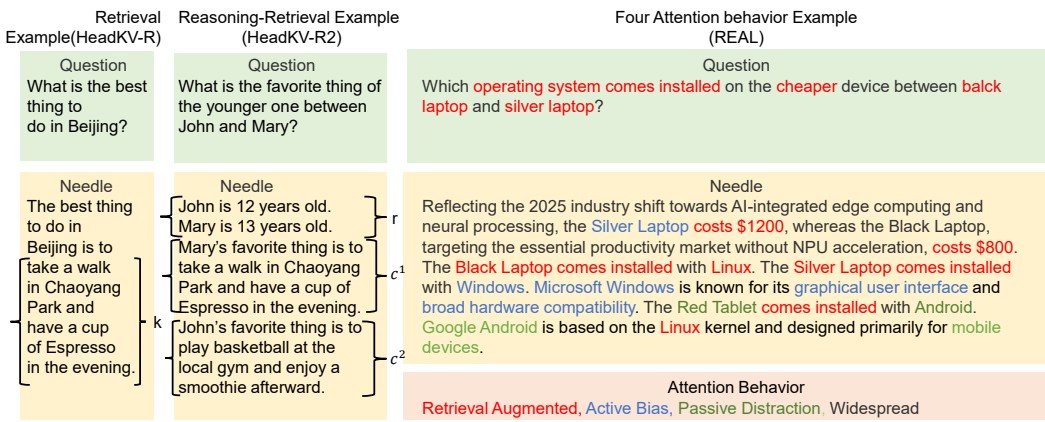

Figure 6: Comparison of needle example between head-wise methods. For HeadKV-R (Fu et al., 2025), the correct answer is directly retrieved from the k part. For HeadKV-R2 (Fu et al., 2025), the correct answer is derived from $c^2$ with background r, ignoring the misleading $c^1$. For REAL, the correct answer is obtained through retrieval-augmented behavior, while effectively mitigating the influence of active bias, passive distraction, and widespread attention. More cases can be seen in Appendix E.

### 2.2 DEFINITION FOR FOUR ATTENTION BEHAVIORS

The essence of previous retrieval-identification methods is shown by the green 'copy-paste retrieval' path in Figure 1, corresponding to useful signals. We argue that the noise should also be modeled to degrade erroneous thought patterns in Figure 6. Widespread behavior simulates the pervasive background context, encompassing unrelated information to reasoning. Retrieval Augmented behavior is the source of the answer, used to construct the logical chain. Active bias behavior is related to the question, but not the source of the answer. Passive distraction behavior is not related to the question and is also not the source of the answer. Active bias and passive distraction share a similar sentence structure and similar surrounding words from the retrieval augmented source. Excessive attention to passive distraction suggests the model is hallucinating or being misled by structural patterns. Focusing solely on the signal (Retrieval Augmented) while ignoring the noise (Bias, Distraction and widespread) leads to a KV cache compression that is insufficient to maintain reasoning robustness, making the model prone to reasoning errors.

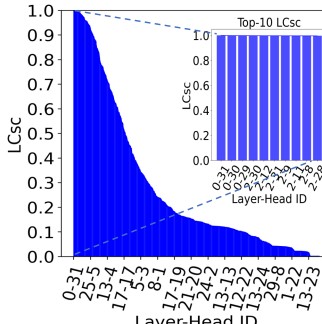 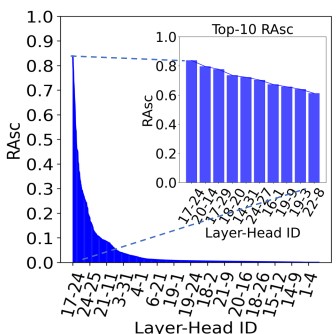 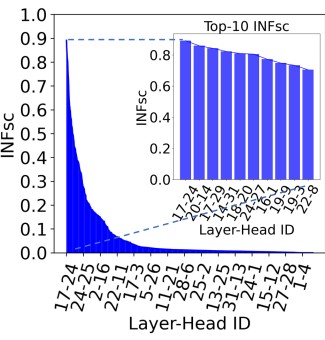

Figure 7: Layer-heads ranked by descending scores, each with a zoomed-in bar chart of the top-10 layer-head IDs. The graphs reveal three key characteristics: i) A small subset of layer-heads have notably higher scores, while most exhibit low values. ii) The distributions vary considerably across layer-heads, indicating that not all heads are equally important. iii) LCsc activates more heads overall, suggesting that bias is widespread but varies in degree.

## 2.3 INFERENCE SCORE CALCULATION

As shown in Equation 1, the attention mechanism serves as the basis for the Attention Weight Confusion Matrix (AWCM).

$$\text{Attention}(Q, K, V) = \text{Softmax}\left(\frac{QK^\top}{\sqrt{d_k}}\right)V \tag{1}$$

Drawing an analogy to precision, recall, and F1-score in confusion matrices (Susmaga, 2004), we introduce the *retrieval-augmented score (RAsc)*, *logic-constructed score (LCsc)*, and *inference score (INFsc)*, as defined in Eq. (2), where RAW, DW, and BW refer to the retrieval-augmented, distracted, and biased weights, respectively, as illustrated in the upper part of Figure 3. On the Llama-3-8B-Instruct model (Grattafiori et al., 2024), the heatmap distributions of these three scores are shown in Figure 4, the relationship between RAsc and LCsc in Figure 5, and the descending score ranking of layer-heads in Figure 7.

$$RAsc = \frac{\text{RAW}}{\text{RAW} + \text{DW}}, \quad LCsc = \frac{\text{RAW}}{\text{RAW} + \text{BW}}, \quad INFsc = \frac{2 \cdot RAsc \cdot LCsc}{RAsc + LCsc} \tag{2}$$

High inference score (INFsc) is achieved when both RAsc and LCsc are simultaneously high. This means the RAW is high, while the DW and BW are low. In this state, the head focuses more on retrieval-augmented behavior and pays less attention to active bias and passive distraction behaviors, and should therefore be allocated a larger KV budget. Low inference score (INFsc) is achieved when both RAsc and LCsc are simultaneously low. This occurs when RAW is low, but DW and BW are high. In this state, the head is paying more attention to active bias and passive distraction behaviors and less attention to retrieval-augmented behavior, thus it should be allocated a smaller KV budget.

## 2.4 LAYER- AND HEAD-WISE KV BUDGET ALLOCATION

Each attention head is initially assigned a fixed KV cache size $b$, which is the KV Size used for the experiments. A fraction of this budget, determined by a predefined ratio $\beta$, is contributed to a shared pool B, leaving a basic budget $b_{\text{base}}$ for each head. A smaller value $\beta$ represents a larger shared budget pool B, meaning that KV cache allocation relies more heavily on inference score (INFsc). The final budget for a head $b_h$ can be seen from Eq. (3), where $L$ and $H$ are the total number of layers and heads in the model, respectively. $L_i$ represents the relative inference score (INFsc) of the whole model, and $S_{i,j}$ represents the relative inference score (INFsc) proportion within $L_i$. All other variables can be derived from these two hyperparameters. Furthermore, it should be noted that the variable $b$ is the optimization target, and strictly speaking, does not count as a hyperparameter.

$$b_{\text{base}} = b - \frac{b}{\beta}, \quad B = \frac{b}{\beta} \cdot L \cdot H, \quad b_h = b_{\text{base}} + B \cdot (\gamma + L_i) \cdot S_{i,j} \tag{3}$$

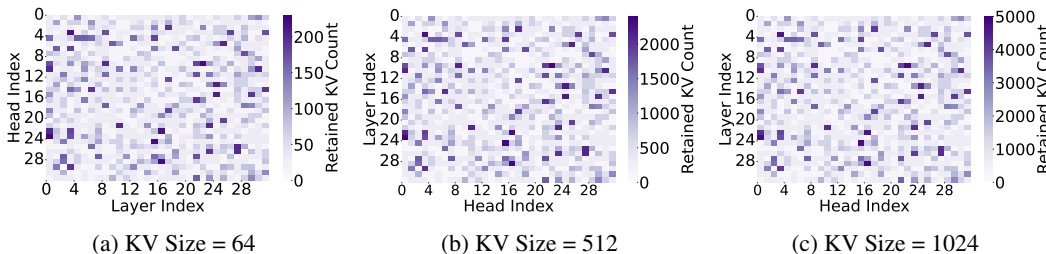

(a) KV Size = 64        (b) KV Size = 512        (c) KV Size = 1024

Figure 8: Retained KV counts under varying KV sizes for Llama-3-8B-Instruct (Grattafiori et al., 2024) on the 18K-length NarrativeQA dataset.

Although the head-wise approach highlights the parallelism of attention heads, it overlooks the hierarchical dependencies across stacked layers (Vaswani et al., 2017), resulting in unstable information transfer. To address this, we empirically assign a base budget $\gamma$ to each layer as a structural safeguard, promoting robust information flow throughout the model. While this setting does not strictly conform to mathematical normalization, it plays a crucial role in preventing over-pruning, which could otherwise impair reasoning.

## 2.5 KV Selection with Allocated Budget

For a given head, if the sequence length remains within budget, all KV states are retained. When the KV length exceeds the limit, the KV states from the most recent query window are first preserved to maintain generation coherence. Attention relevance scores (Li et al., 2024b; Cai et al., 2024; Feng et al., 2024) are then computed between the latest window and earlier cache entries, and only the top-ranked entries that fit within the remaining budget are retained, as follows.

$$S(Q_w, K_c) = \text{Softmax}\left(\frac{Q_w K_c^T}{\sqrt{d_k}}\right), \quad KV_{\text{res}} = \text{Gather}(K_c, \text{TopK}(S, b)) + KV_w \quad (4)$$

In Eq. (4), $S$ is the attention score matrix computed from the query window $Q_w$ and cached keys $K_c$. The parameter $b$ denotes the budget. $KV_{\text{res}}$ represents the compressed KV cache, composed of the fully retained KV states from the latest query window $KV_w$ and the top-ranked entries selected from the previous cache. The detailed algorithm is provided in Appendix B. Figure 8 provides an example of the resulting KV budget distribution across layer-heads.

## 3 Experiments and Analysis

We conducted experiments on three models with maximum context lengths ranging from 8K to 128K: Llama-3-8B-Instruct (Grattafiori et al., 2024), Mistral-7B-Instruct (Jiang et al., 2023), and 123B Mistral-Large-Instruct-2411 (Mistral AI, 2024). Evaluations were performed using three benchmarks, including LooGLE (Li et al., 2024a), LongBench (Bai et al., 2024a) and LongBench v2 (Bai et al., 2024b). LongBench covers both QA and non-QA settings, comprising 16 long-context, knowledge-intensive subsets across six tasks: multi-document QA, single-document QA, summarization, few-shot learning, synthetic reasoning, and code, with sequence lengths from 1K to 18K tokens. LongBench v2 extends this to 20 subsets across six tasks, covering long in-text learning, long-dialogue history understanding, and long structured data understanding, with sequence lengths from 14K to 167K tokens. For evaluation, we selected 27 datasets from these two benchmarks, along with four QA tasks from LooGLE.

Baselines were drawn from different model dimensions: PyramidKV (Cai et al., 2024) and SnapKV (Li et al., 2024b), HeadKV-R (Fu et al., 2025), and HeadKV-R2 (Fu et al., 2025), all under the same KV size for fair comparison. Illustrations of these baselines are provided in Appendix A. For KV entry selection within each head, we set the local window size to $\alpha = 8$ across all five methods. The two introduced hyperparameters, $\beta$ and $\gamma$, are analyzed in Section 3.3.

## 3.1 Main Results

We first present the primary experimental results demonstrating the effectiveness of REAL across QA and non-QA tasks with long contexts.

Table 1: REAL achieves more robust performance at a fixed $\beta$=1.351 across different models and KV size settings. Results for Llama-3-8B-Instruct and Mistral-7B-Instruct are taken from Fu et al. (2025), where $\beta$ fluctuates (e.g., 2.00, 1.20, 1.50). The stability of hyperparameter indicates that the inherent reliability of REAL's attention behavior analysis significantly reduces its dependency on tuning, which greatly benefits the deployment of KV cache compression in real-world scenarios. However, HeadKV (Fu et al., 2025) requires extensive time for parameter tuning across these switching scenarios, which fundamentally introduces external time and influences efficiency. Comprehensive detailed results are in Appendix D. Corresponding compression ratio results are in Appendix H and I.

| Method | Single-Doc QA | | | Multi-Doc QA | | | Avg. | Long Dependency QA | | | | Avg. | $\beta$ |
|---|---|---|---|---|---|---|---|---|---|---|---|---|---|
| | NartQA | Qasper | MF-en | HotpotQA | 2WikiMQA | Musique | | DocQA | Info. Retrieval | Timeline | Computation | | |
| **Llama-3-8B-Instruct, KV Size = Full** | | | | | | | | | | | | | |
| FullKV | 25.56 | 32.07 | 39.71 | 43.57 | 35.28 | 21.18 | 32.90 | 8.73 | 11.21 | 0.67 | 7.43 | 7.01 | - |
| **Llama-3-8B-Instruct, KV Size = 64** | | | | | | | | | | | | | |
| PyramidKV | 21.17 | 13.66 | 29.34 | 34.86 | 23.46 | 15.88 | 23.06 | 8.27 | 9.31 | 0.63 | 6.86 | 6.27 | - |
| SnapKV | 20.51 | 12.80 | 31.69 | 37.02 | 25.91 | 17.02 | 24.16 | 8.84 | 9.43 | **0.66** | 6.18 | 6.28 | - |
| HeadKV-R | 22.67 | 23.54 | 37.51 | 37.45 | 29.76 | 19.01 | 28.32 | 8.80 | 10.51 | 0.58 | 6.68 | 6.64 | 2.00 |
| HeadKV-R2 | 23.21 | 25.33 | **38.71** | 40.64 | **31.33** | 19.35 | 29.76 | **9.46** | 10.66 | 0.61 | 6.92 | 6.91 | 1.20 |
| REAL | **26.22** | **26.30** | 38.05 | **43.89** | 31.06 | **20.75** | **31.05** | 9.23 | **10.67** | 0.63 | **7.42** | **6.99** | 1.351 |
| **Mistral-7B-Instruct, KV Size = Full** | | | | | | | | | | | | | |
| FullKV | 26.63 | 32.99 | 49.34 | 42.77 | 27.35 | 18.78 | 32.98 | 12.17 | 15.52 | 0.49 | 10.03 | 9.55 | - |
| **Mistral-7B-Instruct, KV Size = 64** | | | | | | | | | | | | | |
| PyramidKV | 20.91 | 19.61 | 38.05 | 32.18 | 22.87 | 15.26 | 24.81 | 10.64 | 11.69 | 0.56 | 9.06 | 7.99 | - |
| SnapKV | 19.95 | 18.63 | 38.16 | 31.24 | 21.39 | 13.81 | 23.86 | 10.41 | 11.49 | 0.46 | 9.38 | 7.94 | - |
| HeadKV-R | 24.23 | 25.22 | 46.02 | 38.82 | 26.05 | 17.41 | 29.63 | 10.94 | 13.14 | 0.63 | 9.11 | 8.46 | 1.50 |
| HeadKV-R2 | 21.77 | 26.57 | **48.39** | 40.12 | **26.76** | 16.21 | 29.97 | 11.19 | **13.94** | 0.48 | 9.87 | 8.87 | 1.20 |
| REAL | **24.34** | **27.45** | 47.31 | 40.04 | 25.14 | 17.34 | **30.27** | **12.47** | 13.48 | **0.66** | **10.17** | **9.20** | 1.351 |
| **123B Mistral-Large-Instruct-2411, KV Size = Full** | | | | | | | | | | | | | |
| FullKV | 26.67 | 34.15 | 51.35 | 47.37 | 38.89 | 28.57 | 37.83 | 16.67 | 18.18 | 1.94 | 16.76 | 13.39 | - |
| **123B Mistral-Large-Instruct-2411, KV Size = 64** | | | | | | | | | | | | | |
| PyramidKV | 20.75 | 21.21 | 41.49 | 31.58 | 21.95 | 13.64 | 25.10 | 13.83 | 11.58 | 0.57 | 9.18 | 8.79 | - |
| SnapKV | 19.51 | 21.95 | 39.36 | 33.33 | 21.21 | 14.29 | 24.94 | 13.33 | 12.05 | 0.54 | 9.09 | 8.76 | - |
| HeadKV-R | 23.33 | 23.62 | 43.25 | 31.71 | 23.81 | 14.00 | 26.62 | 14.24 | 15.11 | 0.63 | 12.33 | 10.58 | 1.351 |
| HeadKV-R2 | 22.73 | 24.39 | 45.95 | 35.14 | **27.84** | 14.43 | 28.41 | 14.68 | 16.84 | **0.83** | 12.86 | 11.30 | 1.351 |
| REAL | **26.83** | **27.28** | **47.37** | **38.10** | 27.27 | 18.33 | **30.86** | **15.17** | **16.26** | 0.77 | **13.20** | **11.35** | 1.351 |

Table 2: REAL outperforms baselines on Non-QA tasks. Results for the first two models are from Fu et al. (2025).

| Method | Summarization | | | Few-Shot Learning | | | Synthetic | | Code | | Avg. |
|---|---|---|---|---|---|---|---|---|---|---|---|
| | GovReport | QMSum | MultiNews | TREC | TriviaQA | SAMSum | PCount | PRe | LCC | RB-P | |
| **Llama-3-8B-Instruct, KV Size = Full** | | | | | | | | | | | |
| FullKV | 28.71 | 23.26 | 26.64 | 73.50 | 90.48 | 42.33 | 4.80 | 69.25 | 59.29 | 54.05 | 47.23 |
| **Llama-3-8B-Instruct, KV Size = 128** | | | | | | | | | | | |
| PyramidKV | 20.16 | 20.06 | 21.34 | 66.53 | 89.63 | 37.87 | 5.08 | 69.45 | 59.58 | 58.44 | 44.82 |
| SnapKV | 19.83 | 21.80 | 21.41 | 65.50 | 89.72 | 38.71 | 5.75 | 69.00 | 58.74 | 54.57 | 44.50 |
| HeadKV-R | 21.08 | 22.35 | 22.50 | 71.50 | 89.45 | 38.40 | 5.00 | **69.50** | 60.89 | 59.92 | 46.06 |
| HeadKV-R2 | 21.76 | 22.16 | 23.94 | 71.50 | 90.19 | 38.88 | **6.60** | 69.50 | **61.08** | **60.21** | 46.58 |
| REAL | **23.34** | **23.19** | **24.76** | **75.28** | **90.57** | 40.37 | 5.23 | 69.50 | 60.37 | 59.72 | **47.23** |
| **Mistral-7B-Instruct, KV Size = Full** | | | | | | | | | | | |
| FullKV | 32.87 | 24.24 | 27.10 | 71.00 | 86.23 | 42.79 | 2.75 | 86.98 | 56.93 | 54.49 | 48.54 |
| **Mistral-7B-Instruct, KV Size = 128** | | | | | | | | | | | |
| PyramidKV | 21.23 | 22.16 | 21.80 | 68.69 | 86.53 | 40.77 | 3.58 | 70.20 | 53.57 | 49.02 | 43.76 |
| SnapKV | 20.76 | 22.72 | 21.38 | 67.00 | 85.06 | 40.22 | 3.51 | 65.06 | 52.20 | 47.01 | 42.49 |
| HeadKV-R | 22.19 | 22.86 | 22.57 | 69.50 | 85.46 | 41.16 | 3.56 | 74.49 | 54.60 | 50.89 | 44.73 |
| HeadKV-R2 | 24.30 | 23.48 | 24.18 | 70.50 | 85.54 | 40.72 | **4.83** | 72.63 | **55.49** | **51.39** | 45.31 |
| REAL | **28.23** | 23.64 | 26.36 | **73.53** | **86.55** | 42.58 | 3.06 | **83.58** | 54.34 | 51.17 | **47.30** |
| **123B Mistral-Large-Instruct-2411, KV Size = Full** | | | | | | | | | | | |
| FullKV | 36.58 | 27.77 | 27.84 | 75.53 | 94.44 | 44.79 | 5.19 | 93.81 | 59.50 | 58.39 | 52.38 |
| **123B Mistral-Large-Instruct-2411, KV Size = 128** | | | | | | | | | | | |
| PyramidKV | 23.40 | 25.34 | 24.67 | 70.66 | 89.77 | 38.81 | 5.24 | 84.96 | 53.32 | 54.29 | 47.04 |
| SnapKV | 22.78 | 23.49 | 24.61 | 69.14 | 91.64 | 39.25 | 5.01 | 81.17 | 54.38 | 56.32 | 46.78 |
| HeadKV-R | 25.83 | 27.09 | 24.75 | 71.59 | 90.88 | 40.06 | 5.33 | 88.91 | 56.26 | 56.44 | 48.71 |
| HeadKV-R2 | 27.32 | 27.18 | 25.52 | 72.10 | 92.15 | 41.69 | **6.00** | 87.15 | 57.18 | 57.20 | 49.34 |
| REAL | **27.92** | **27.74** | **26.76** | **74.18** | **93.53** | **43.24** | 5.65 | **90.89** | **58.49** | **57.70** | **50.61** |

**LongBench Evaluation across QA and Non-QA Tasks (1K–18K Tokens)** QA results are shown in Table 1 and non-QA results in Table 9. Across all tasks, REAL outperforms the four baselines

Table 3: REAL achieves superior performance on LongBench v2 (Bai et al., 2024b) using the 123B Mistral-Large-Instruct-2411 model (Mistral AI, 2024).

| Method | Single-Doc QA | | | Multi-Doc QA | | | In-context | | Dialogue | Struct | | Avg. |
|---|---|---|---|---|---|---|---|---|---|---|---|---|
| | Literary | Legal | Detective | Academic | Financial | Govern | UserGuide | Many-shot | DialogueHistory | Table | KnGraph | |
| **Mistral-Large-Instruct-2411, KV Size = Full** | | | | | | | | | | | | |
| FullKV | 33.34 | 30.29 | 18.08 | 37.23 | 43.24 | 29.27 | 34.98 | 42.86 | 42.11 | 27.74 | 40.00 | 34.47 |
| **Mistral-Large-Instruct-2411, KV Size = 64** | | | | | | | | | | | | |
| PyramidKV | 20.03 | 18.89 | 3.64 | 24.76 | 32.50 | 20.73 | 23.81 | 13.64 | 13.51 | 8.11 | 16.67 | 17.84 |
| SnapKV | 21.19 | 16.22 | 3.86 | 23.81 | 31.71 | 19.98 | 22.22 | 13.51 | 14.63 | 9.52 | 13.64 | 17.31 |
| HeadKV-R | 24.11 | 19.91 | 4.90 | 26.57 | 35.08 | 21.84 | 24.13 | 15.60 | 18.69 | 10.53 | 17.32 | 19.83 |
| HeadKV-R2 | 25.48 | 17.12 | 6.84 | 27.50 | 35.21 | 23.25 | 25.86 | 21.50 | 22.17 | 10.81 | 20.98 | 21.52 |
| REAL | **26.45** | **20.92** | **8.70** | **29.27** | **36.42** | **23.50** | **28.57** | **22.51** | **24.01** | **14.21** | **20.99** | **23.23** |
| **Mistral-Large-Instruct-2411, KV Size = 128** | | | | | | | | | | | | |
| PyramidKV | 22.94 | 18.23 | 5.10 | 31.58 | 35.08 | 23.54 | 25.26 | 15.60 | 16.22 | 4.55 | 16.67 | 19.52 |
| SnapKV | 23.93 | 16.67 | 5.25 | 32.05 | 33.51 | 22.46 | 25.05 | 16.22 | 14.06 | 9.09 | 13.33 | 19.24 |
| HeadKV-R | 24.23 | 18.04 | 7.89 | 33.35 | **36.36** | 23.79 | 26.87 | 17.91 | 20.35 | 13.24 | 19.84 | 21.99 |
| HeadKV-R2 | 25.48 | 18.12 | 9.52 | **33.82** | 35.11 | 24.52 | 26.50 | 24.27 | 24.80 | 14.29 | 21.22 | 23.42 |
| REAL | **27.28** | **22.61** | **12.41** | 33.76 | 36.28 | **25.54** | **28.91** | **25.77** | **26.83** | **16.43** | **25.03** | **25.53** |
| **Mistral-Large-Instruct-2411, KV Size = 256** | | | | | | | | | | | | |
| PyramidKV | 26.67 | 21.74 | 4.12 | 35.11 | 36.81 | 22.22 | 27.14 | 18.15 | 26.52 | 10.33 | 22.10 | 22.81 |
| SnapKV | 23.36 | 18.87 | 4.55 | 36.17 | 37.84 | 24.39 | 27.50 | 19.05 | 26.32 | 11.11 | 20.39 | 22.69 |
| HeadKV-R | 27.42 | 22.35 | 7.87 | 36.35 | 37.85 | 25.60 | 29.27 | 23.52 | 30.70 | 16.67 | 23.64 | 25.57 |
| HeadKV-R2 | 29.68 | 23.81 | 9.05 | **36.84** | **39.12** | **28.84** | **31.84** | 30.50 | 32.36 | 19.05 | 25.38 | 27.86 |
| REAL | **32.11** | **25.42** | **13.64** | 36.26 | 39.01 | 28.74 | 31.56 | **33.69** | **35.15** | **21.18** | **27.29** | **29.64** |

Table 4: Ablation study without Retrieval-Augmented, Distracted, and Biased Weights.

| Method | Single-Doc QA | | | Multi-Doc QA | | | Long Dependency QA | | | | | Avg. |
|---|---|---|---|---|---|---|---|---|---|---|---|---|
| | NartQA | Qasper | MF-en | HotpotQA | 2WikiMQA | Musique | DocQA | Info. Retrieval | Timeline | Computation | | |
| **Llama-3-8B-Instruct, KV Size = 64** | | | | | | | | | | | | |
| +RAW, –DW, –BW | 25.94 | 23.93 | 35.79 | 41.57 | 29.93 | **21.07** | 8.61 | **10.91** | 0.60 | 7.11 | | 20.55 |
| +RAW, +DW, –BW | 22.29 | 11.08 | 22.93 | 29.78 | 18.16 | 12.19 | 6.89 | 7.09 | 0.00 | 4.88 | | 13.53 |
| +RAW, –DW, +BW | 22.26 | 14.81 | 22.73 | 26.02 | 18.78 | 12.82 | 6.90 | 7.12 | 0.00 | 4.85 | | 13.63 |
| -RAW, +DW, +BW | 23.97 | 14.55 | 34.74 | 42.96 | **35.44** | 21.19 | 9.18 | 10.33 | 0.54 | 6.57 | | 19.95 |
| +RAW, +DW, +BW | **26.22** | **26.30** | **38.05** | **43.89** | 31.06 | 20.75 | **9.23** | 10.67 | **0.63** | **7.42** | | **21.42** |
| **Llama-3-8B-Instruct, KV Size = 128** | | | | | | | | | | | | |
| +RAW, –DW, –BW | 25.92 | 28.71 | 36.82 | 44.46 | 32.28 | 20.85 | 9.04 | 11.16 | **0.62** | 7.40 | | 7.06 |
| +RAW, +DW, –BW | 22.39 | 17.43 | 32.28 | 39.94 | 27.26 | 18.10 | 9.15 | 9.64 | 0.52 | 7.54 | | 6.71 |
| +RAW, –DW, +BW | 24.87 | 30.22 | **38.00** | 40.04 | 26.02 | 18.38 | 8.71 | 10.08 | 0.50 | 7.21 | | 6.63 |
| -RAW, +DW, +BW | **26.14** | 26.92 | 37.46 | 43.74 | **36.56** | 20.40 | **9.58** | 11.06 | 0.45 | 7.58 | | 7.17 |
| +RAW, +DW, +BW | 25.47 | **29.95** | 38.02 | 44.67 | 34.28 | **20.66** | 9.20 | **11.32** | **0.62** | **7.83** | | 7.24 |

Table 5: Results for QA tasks on Llama-3-8B-Instruct and Mistral-7B-Instruct models with different KV cache sizes. Additional baselines include `Random_Head`, `Mean_Attn_Score`, and `Max_Attn_Score`. HeadKV-R and HeadKV-R2 results are from Fu et al. (2025).

| Method | Single-Doc QA | | | Multi-Doc QA | | | Avg. | Long Dependency QA | | | | Avg. |
|---|---|---|---|---|---|---|---|---|---|---|---|---|
| | NartQA | Qasper | MF-en | HotpotQA | 2WikiMQA | Musique | | DocQA | Info. Retrieval | Timeline | Computation | |
| **Llama-3-8B-Instruct, KV Size = 64** | | | | | | | | | | | | |
| FullKV | 25.56 | 32.07 | 39.71 | 43.57 | 35.28 | 21.18 | 32.90 | 8.73 | 11.21 | 0.67 | 7.43 | 7.01 |
| Random_Head | 15.20 | 10.50 | 25.10 | 22.30 | 15.40 | 10.20 | 16.45 | 5.00 | 5.50 | 0.30 | 3.50 | 3.58 |
| Mean_Attn_Score | 19.80 | 14.20 | 30.50 | 28.40 | 20.10 | 13.50 | 21.08 | 6.00 | 6.80 | 0.40 | 4.50 | 4.43 |
| Max_Attn_Score | 21.50 | 18.40 | 33.20 | 32.15 | 24.50 | 15.80 | 24.26 | 7.50 | 8.20 | 0.50 | 5.80 | 5.50 |
| HeadKV-R | 22.67 | 23.54 | 37.51 | 37.45 | 29.76 | 19.01 | 28.32 | 8.80 | 10.51 | 0.58 | 6.68 | 6.64 |
| HeadKV-R2 | 23.21 | 25.33 | **38.71** | 40.64 | **31.33** | 19.35 | 29.76 | **9.46** | 10.66 | 0.61 | 6.92 | 6.91 |
| REAL | **26.22** | **26.30** | 38.05 | **43.89** | 31.06 | **20.75** | **31.05** | 9.23 | **10.67** | **0.63** | **7.42** | **6.99** |
| **Mistral-7B-Instruct, KV Size = 64** | | | | | | | | | | | | |
| FullKV | 26.63 | 32.99 | 49.34 | 42.77 | 27.35 | 18.78 | 32.98 | 12.17 | 15.52 | 0.49 | 10.03 | 9.55 |
| Random_Head | 14.20 | 12.50 | 28.10 | 24.30 | 15.20 | 9.80 | 17.35 | 6.10 | 6.80 | 0.25 | 4.80 | 4.49 |
| Mean_Attn_Score | 18.50 | 16.20 | 35.80 | 29.40 | 19.50 | 12.00 | 21.90 | 8.20 | 8.50 | 0.35 | 6.50 | 5.89 |
| Max_Attn_Score | 22.50 | 21.00 | 40.50 | 33.50 | 23.10 | 14.50 | 25.85 | 10.50 | 10.20 | 0.42 | 8.20 | 7.33 |
| HeadKV-R | 24.23 | 25.22 | 46.02 | 38.82 | 26.05 | **17.41** | 29.63 | 13.14 | 13.14 | 0.63 | 9.11 | 8.46 |
| HeadKV-R2 | 21.77 | 26.57 | **48.39** | **40.12** | **26.76** | 16.21 | 29.97 | 13.94 | **13.94** | 0.48 | 9.87 | 8.87 |
| REAL | **24.34** | **27.45** | 47.31 | 40.04 | 25.14 | 17.34 | **30.27** | 13.48 | 13.48 | **0.66** | **10.17** | **9.20** |

under the same KV cache budget, underscoring the effectiveness of proposed comprehensive metric inference score (INFsc). Notably, REAL even surpasses FullKV on the TREC dataset (with KV size = 128 on Llama-3-8B-Instruct (Grattafiori et al., 2024)) and the PRe dataset (with KV size = 128 on Mistral-7B-Instruct (Jiang et al., 2023)). Comprehensive results are provided in Appendix

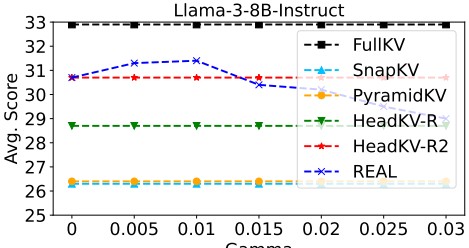
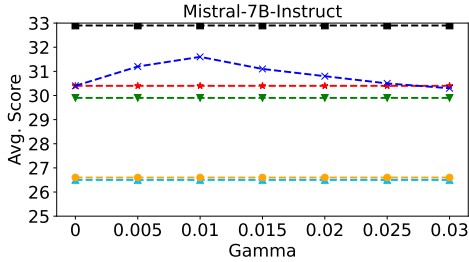

Figure 9: Average accuracy across six LongBench datasets (Bai et al., 2024a) for different $\gamma$ values. At a constrained KV size of 64, the scarcity of resources more clearly reveals the minimal budget required by each layer.

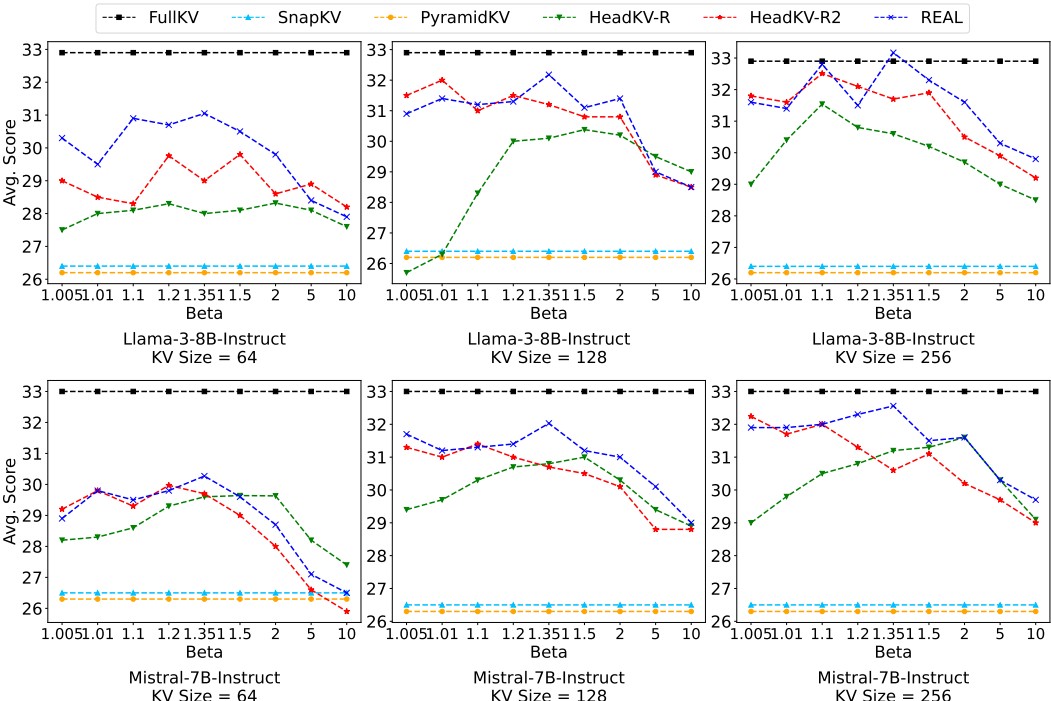

Figure 10: Average scores across LongBench datasets (Bai et al., 2024a) for different values of $\beta$.

D. Furthermore, REAL achieves stable performance under robust $\beta$, unlike the instability reported in HeadKV-R (Fu et al., 2025) and HeadKV-R2 (Fu et al., 2025).

**LongBench v2 Evaluation across QA and Non-QA Tasks (14K-129K Tokens)** Table 3 further demonstrates the effectiveness of REAL on much longer sequences, which require the construction of long-range logical dependencies. In contrast to typical QA tasks, non-QA tasks demand that models move beyond surface-level interpretation to capture implicit meanings in context.

## 3.2 ABLATION STUDY

We conduct two kinds of baselines. One is attention behavior ablation. The other is head selection ablation. For attention behavior, we investigate retrieval-augmented, distracted, biased and widespread behavior. Table 4 shows that retrieval-augmented behavior achieves the best performance on the Musique and Info. Retrieval datasets even with a KV size as small as 64, indicating that the model naturally relies on retrieval mechanisms under resource-constrained settings. When the KV cache size increases slightly (e.g., to 128), the model gains more cognitive space to achieve better performance. Notably, at KV sizes of 64 and 128, distracted and biased analysis yields more pronounced effects on the 2WikiMQA dataset. For head selection, we add Random_Head,

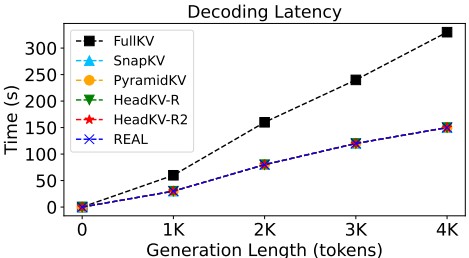 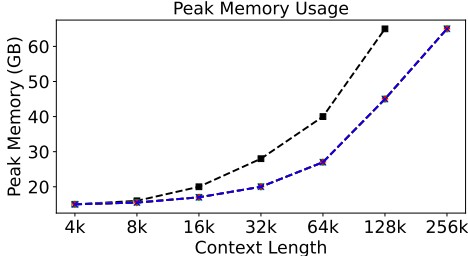

Figure 11: Computation efficiency of REAL compared with other KV cache compression baselines, measured by decoding latency and peak memory usage. When measuring the average one token generation time, the results show that FullKV is the fastest at 4.45s, followed by REAL (4.63s), HeadKV-R2 (4.72s), HeadKV-R (4.86s), PyramidKV (5.02s), and SnapKV (5.07s).

Max_Attention_Score_Head, Mean_Attention_Score_Head. Table 5 shows partial results. The detailed description is in Appendix G.

### 3.3 Hyperparameter Analysis

REAL only introduces two hyperparameters: $\gamma$, which enforces a minimum per-layer budget to preserve essential representations, and $\beta$, which controls the size of the global shared budget pool $B$. We selected $\gamma \in \{0, 0.005, 0.01, 0.015, 0.02, 0.025, 0.03\}$ and $\beta \in \{1.005, 1.01, 1.1, 1.2, 1.351, 1.5, 2, 5, 10\}$, as ahown in Figure 9 and Figure 10. The combined effect of well-chosen hyperparameters enhances overall performance.

## 4 Peak Memory and Decoding Latency

We evaluate the computational efficiency of REAL on the Mistral-7B-Instruct model (Jiang et al., 2023) with a maximum sequence length of 32K. Decoding latency is measured using 32K-length inputs from the Reasoning-in-a-Haystack dataset (Kuratov et al., 2024) with generation lengths of 1, 512, 1024, 2048, and 4096.

During the prefilling stage, decoding latency includes both the prefilling time and the decoding time. After the model finishes encoding the input sample, the prefilling stage performs KV cache compression. After prefilling is complete, the decoding stage outputs tokens one by one. When the generation length is set to 1, the overhead of the generation itself is almost negligible. The decoding latency is approximately equal to the prefilling time. As shown in Figure 11, REAL attains latency comparable to other KV compression baselines while remaining closest in performance to the FullKV. We also measured peak memory usage during decoding, averaged over ten runs, as shown in Figure 11. Compared with FullKV, REAL significantly reduces memory usage while maintaining performance on par with other baselines.

## 5 Conclusion

In this paper, we propose a novel and robust two-stage REtrieval-Augmented and Logic-constructed (REAL) method for KV cache compression at layer-head granularity. In the first stage, attention behaviors are analyzed using the attention weight confusion matrix (AWCM), which categorizes weights into retrieval-augmented, distracted, biased, and widespread behaviors with a designed needle in Needle-in-a-Haystack (Kamradt, 2023). In the second stage, Inference score (INFsc), defined as the harmonic mean of the retrieval-augmented score (RAsc) and logic-constructed score (LCsc), guides dynamic KV budget allocation in addition to a basic budget. To ensure robust information transfer across stacked layers, we further assign an empirical base budget of $\gamma$ per layer.

Evaluations on multiple tasks from LongBench (Bai et al., 2024a), LongBench v2 (Bai et al., 2024b), and LooGLE (Li et al., 2024a) demonstrate that REAL outperforms token-, layer-, and head-wise baselines. Moreover, REAL achieves lower decoding latency and memory usage, highlighting its practicality for real-world deployment.

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

## A KV Cache compression illustration

Figures 12 to 14 and 16 show the illustration of other KV cache compression methods.

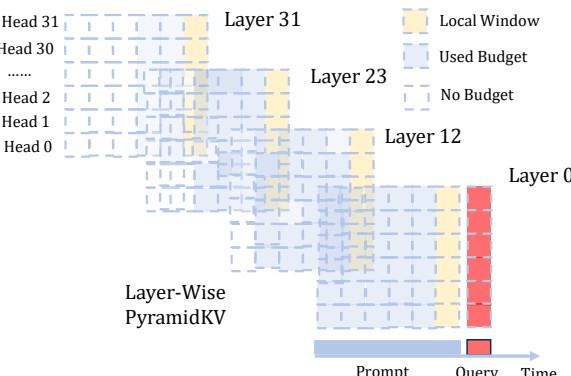

Figure 12: PyramidKV compresses the KV cache by allocating more cache in lower layers and less in higher ones.

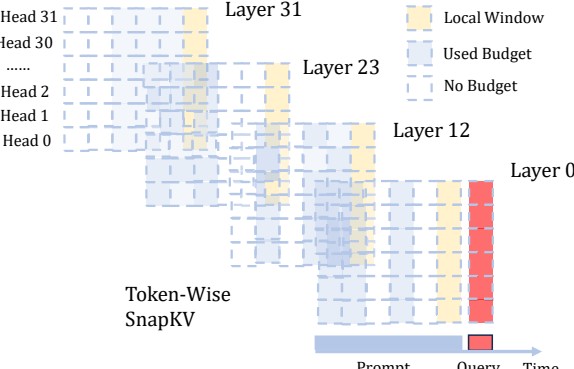

Figure 13: SnapKV compresses the KV cache by selecting grouped important positions for each attention head.

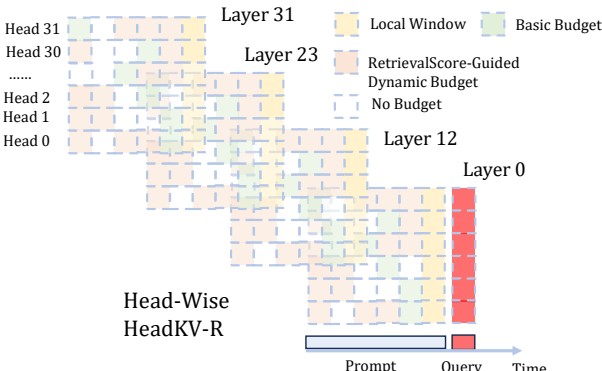

Figure 14: HeadKV-R compresses the KV cache by focusing on the retrieval-and-paste mechanism for each head.

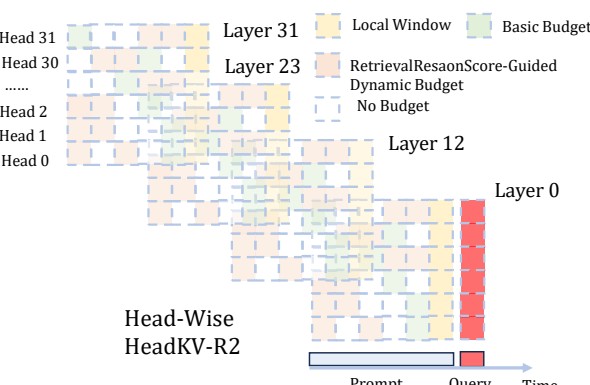

Figure 15: HeadKV-R2 compresses the KV cache by considering the contextual and reasoning skills for each head.

## B KV CACHE BUDGET ALLOCATION

Algorithms 1 and 2 present the REAL method for KV cache compression by selecting the top-$k$ entries. The procedure first checks whether the KV length exceeds the budget. If it does not, the original $K$ and $V$ are returned unchanged. If it does, the full query window is retained, and the top-$k$ most relevant historical tokens within the budget are selected to form the compressed KV.

---

**Algorithm 1** Procedure for KV budget allocation.

---

**Input**: Total budget $b_c$, allocation ratio $\beta$, head $(i, j)$, layer count $L$, head count in a layer $H$, INFsc of $j_{\text{th}}$ head$(i, j)$ in the $i_{\text{th}}$ layer $H_{j,\text{inf}}$
**Output**: Capacity C$i, j$

1: Basic budget $b_{\text{base}} = b\left(1 - \frac{1}{\beta}\right)$
2: Global dynamic budget pool $B_{\text{total}} = \frac{b}{\beta} L H$
3: $i_{th}$ layer's total INFScore $L_{i,\alpha} = \sum_j H_{j,\text{inf}}$
4: $L_s \leftarrow \sum_{i=0}^{L-1} L_{i,\alpha}$,
5: Relative Importance of head and layer $\quad S_{j,h} \leftarrow \frac{H_{j,\text{inf}}}{L_s}, L_{i,h} = \frac{L_{i,\alpha}}{L_s}$
6: Dynamic allocation $b_{i,j}^{\text{dyn}} = B_{\text{total}} \cdot (\gamma + L_{i,h}) \cdot S_{j,h}$
7: Total KV cache budget $b_{i,j} = b_{\text{base}} + b_{i,j}^{\text{dyn}}$
8: **return** C$_{i,j} = \left\lfloor \max(0, b_{i,j}) + 0.5 \right\rfloor$

---

**Algorithm 2** Procedure for KV entry selection.

---

**Input**: key states $K$, query states $Q$, value states $V$, budget $C$
**Output**: Final_KV

1: **if** current_KV_length $\leq B$ **then**
2:     **return** $K, V$
3: **end if**
4: $W \leftarrow$ Last window-size tokens from KV
5: $Q H \leftarrow$ Remaining history from KV
6: Attention relevance $A = \text{Softmax}\left(Q_{\text{window}} H^T / \sqrt{d_k}\right)$
7: Budget for history $C_H = C - \text{window\_size}$
8: TopK_indices $= \text{TopK}(A, k = B_H)$
9: Relevant history $H' = \text{Gather}(H, \text{TopK\_indices})$
10: Final_KV $= \text{Concat}(H', W)$
11: **return** Final_KV

---

## C   DATASET DETAILS

Table 6 shows the details of sixteen QA datasets from LongBench (Bai et al., 2024a), four QA datasets from LooGLE (Li et al., 2024a), and eleven extended-length datasets from LongBench v2 (Bai et al., 2024b).

Table 6: Dataset Details.

| Label | Task | Avg Len |
|---|---|---|
| NrtvQA | NarrativeQA | 18,409 |
| Qasper | Qasper | 3,619 |
| MF-en | MultiFieldQA-EN | 4,559 |
| HotpotQA | HotpotQA | 9,151 |
| 2WikiMultiHopQA | 2WikiMultiHopQA | 4,887 |
| Musique | Musique | 11,214 |
| QMSum | QMSum | 10614 |
| MultiNews | MultiNews | 2113 |
| TREC | TREC | 5177 |
| TriviaQA | TriviaQA | 8209 |
| SAMSum | SAMSum | 6258 |
| PCount | PassageCount | 11141 |
| PRe | PassageRetrieval-en | 9289 |
| LCC | LCC | 1235 |
| RB-P | RepoBench-P | 4206 |
| GovReport | Government report | 8734 |
| Doc.QA | Comprehension & reasoning | 15,498 |
| Info.Retrieval | Multiple information retrieval | 14,808 |
| Timeline | Timeline reorder | 15,425 |
| Computation | Computation | 17,001 |
| Literary | Literary | 72K |
| Legal | Legal | 28K |
| Detective | Detective | 70K |
| Detective | Academic | 27K |
| Financial | Financial | 49K |
| Govern | Government reports | 20K |
| UserGuide | User guide QA | 61K |
| Many-Shot | Many-Shot | 71K |
| Dialogue History | Dialogue history QA | 77K |
| Table | Table QA | 42K |
| KnGraph | Knowledge graph reasoning | 52K |

# D   COMPREHENSIVE EXPERIMENTAL RESULTS ON LONGBENCH

Table 7 shows that the REAL method achieves substantial improvements over the baselines across different settings.

Table 7: Results for varying KV cache sizes (64, 128, 256, 512, 1024) on Llama-3-8B-Instruct and Mistral-7B-Instruct, with baseline results reported by Fu et al. (2025).

| Method | Single-Doc QA | | | Multi-Doc QA | | | Avg. | Long Dependency QA | | | | Avg. | β |
|---|---|---|---|---|---|---|---|---|---|---|---|---|---|
| | NartQA | Qasper | MF-en | HotpotQA | 2WikiMQA | Musique | | DocQA | Info. Retrieval | Timeline | Computation | | |
| **Llama-3-8B-Instruct, KV Size = Full** | | | | | | | | | | | | | |
| FullKV | 25.56 | 32.07 | 39.71 | 43.57 | 35.28 | 21.18 | 32.90 | 8.73 | 11.21 | 0.67 | 7.43 | 7.01 | - |
| **Llama-3-8B-Instruct, KV Size = 64** | | | | | | | | | | | | | |
| PyramidKV | 21.17 | 13.66 | 29.34 | 34.86 | 23.46 | 15.88 | 23.06 | 8.27 | 9.31 | 0.63 | 6.86 | 6.27 | - |
| SnapKV | 20.51 | 12.80 | 31.69 | 37.02 | 25.91 | 17.02 | 24.16 | 8.84 | 9.43 | **0.66** | 6.18 | 6.28 | - |
| HeadKV-R | 22.67 | 23.54 | 37.51 | 37.45 | 29.76 | 19.01 | 28.32 | 8.80 | 10.51 | 0.58 | 6.68 | 6.64 | 2 |
| HeadKV-R2 | 23.21 | 25.33 | **38.71** | 40.64 | **31.33** | 19.35 | 29.76 | **9.46** | 10.66 | 0.61 | 6.92 | 6.91 | 1.2 |
| REAL | **26.22** | **26.30** | 38.05 | **43.89** | 31.06 | **20.75** | **31.05** | 9.23 | **10.67** | 0.63 | **7.42** | **6.99** | 1.351 |
| **Llama-3-8B-Instruct, KV Size = 128** | | | | | | | | | | | | | |
| PyramidKV | 22.01 | 17.05 | 31.52 | 39.27 | 28.99 | 18.34 | 26.20 | 8.89 | 9.63 | 0.61 | 6.72 | 6.46 | - |
| SnapKV | 22.11 | 15.79 | 31.01 | 41.12 | 29.20 | 19.35 | 26.43 | 8.36 | 9.46 | **0.79** | 6.56 | 6.29 | - |
| HeadKV-R | 23.49 | 25.39 | 38.15 | 42.45 | 32.84 | 19.95 | 30.38 | 8.87 | 10.35 | 0.78 | 7.52 | 6.88 | 1.5 |
| HeadKV-R2 | 21.80 | 29.19 | **41.89** | 43.73 | **35.01** | 20.40 | 32.00 | **9.60** | 11.13 | 0.67 | 7.22 | 7.16 | 1.01 |
| REAL | **25.47** | **29.95** | 38.02 | **44.67** | 34.28 | **20.66** | **32.18** | 9.20 | **11.32** | 0.62 | **7.83** | **7.24** | 1.351 |
| **Llama-3-8B-Instruct, KV Size = 256** | | | | | | | | | | | | | |
| PyramidKV | 23.94 | 20.27 | 36.27 | 42.51 | 31.44 | 19.99 | 29.07 | 8.66 | 10.61 | 0.53 | 6.98 | 6.70 | - |
| SnapKV | 23.38 | 20.18 | 37.65 | 42.80 | 33.23 | 20.01 | 29.54 | 9.04 | 10.59 | 0.53 | 7.53 | 6.92 | - |
| HeadKV-R | 23.83 | 29.04 | **39.90** | 42.36 | 33.58 | 20.57 | 31.54 | 9.05 | 11.15 | 0.52 | 7.22 | 6.99 | 1.1 |
| HeadKV-R2 | 24.68 | **30.49** | 38.59 | 44.32 | 36.41 | 20.54 | 32.51 | 9.47 | 11.56 | 0.54 | **7.65** | 7.31 | 1.1 |
| REAL | **26.10** | 30.26 | 37.14 | **44.61** | **38.53** | **22.40** | **33.17** | **10.01** | **11.71** | 0.53 | 7.23 | **7.37** | 1.351 |
| **Llama-3-8B-Instruct, KV Size = 512** | | | | | | | | | | | | | |
| PyramidKV | 24.69 | 23.65 | 35.10 | 43.25 | 31.16 | 20.06 | 29.65 | 8.90 | 10.62 | 0.74 | **7.57** | 6.96 | - |
| SnapKV | 25.47 | 23.75 | 38.64 | 43.66 | 33.98 | 19.83 | 30.89 | 9.00 | 11.07 | 0.63 | 7.34 | 7.01 | - |
| HeadKV-R | 23.84 | 29.21 | **39.79** | 44.41 | 36.09 | 20.59 | 32.32 | 9.13 | 11.61 | **0.56** | 7.12 | 7.11 | 1.2 |
| HeadKV-R2 | 24.75 | 29.75 | 38.03 | **44.43** | 36.45 | 21.67 | 32.51 | **9.34** | 11.26 | **0.56** | 7.54 | 7.18 | 1.1 |
| REAL | **26.13** | **31.58** | 38.94 | 44.23 | **36.76** | **22.82** | **33.41** | 9.32 | **11.97** | 0.55 | 7.34 | **7.30** | 1.351 |
| **Llama-3-8B-Instruct, KV Size = 1024** | | | | | | | | | | | | | |
| PyramidKV | 25.38 | 26.83 | 36.90 | 44.09 | 34.24 | 21.49 | 31.49 | 8.98 | 11.41 | 0.53 | 6.96 | 6.97 | - |
| SnapKV | 25.76 | 27.50 | 38.38 | 43.40 | 34.81 | 20.07 | 31.65 | 9.61 | 11.34 | 0.53 | 7.22 | 7.18 | - |
| HeadKV-R | 24.85 | 30.94 | **39.82** | 43.52 | 36.58 | 20.37 | 32.68 | 9.20 | **11.67** | 0.55 | 7.71 | **7.28** | 1.2 |
| HeadKV-R2 | 24.66 | 30.82 | 39.56 | 43.97 | 36.47 | 22.24 | 32.95 | 9.02 | 11.51 | 0.47 | **7.85** | 7.21 | 1.2 |
| REAL | 25.57 | **32.37** | 39.50 | **44.72** | 36.87 | **22.45** | **33.58** | 9.68 | 11.48 | 0.49 | 7.12 | 7.19 | **1.351** |
| **Mistral-7B-Instruct, KV Size = Full** | | | | | | | | | | | | | |
| FullKV | 26.63 | 32.99 | 49.34 | 42.77 | 27.35 | 18.78 | 32.98 | 12.17 | 15.52 | 0.49 | 10.03 | 9.55 | - |
| **Mistral-7B-Instruct, KV Size = 64** | | | | | | | | | | | | | |
| PyramidKV | 20.91 | 19.61 | 38.05 | 32.18 | 22.87 | 15.26 | 24.81 | 10.64 | 11.69 | 0.56 | 9.06 | 7.99 | - |
| SnapKV | 19.95 | 18.63 | 38.16 | 31.24 | 21.39 | 13.81 | 23.86 | 10.41 | 11.49 | 0.46 | 9.38 | 7.94 | - |
| HeadKV-R | 24.23 | 25.22 | 46.02 | 38.82 | 26.05 | 17.41 | 29.63 | 10.94 | 13.14 | 0.63 | 9.11 | 8.46 | 1.5 |
| HeadKV-R2 | 21.77 | 26.57 | **48.39** | 40.12 | **26.76** | 16.21 | 29.97 | **13.94** | 13.48 | 0.48 | 9.87 | 8.87 | 1.2 |
| REAL | **24.34** | **27.45** | 47.31 | 40.04 | 25.14 | **17.34** | **30.27** | 12.47 | 13.48 | **0.66** | **10.17** | **9.20** | 1.351 |
| **Mistral-7B-Instruct, KV Size = 128** | | | | | | | | | | | | | |
| PyramidKV | 21.76 | 21.98 | 43.72 | 32.76 | 22.73 | 15.59 | 26.42 | 10.64 | 11.90 | 0.47 | 8.69 | 7.93 | - |
| SnapKV | 21.47 | 21.95 | 45.24 | 33.88 | 21.83 | 15.53 | 26.65 | 10.86 | 12.24 | 0.57 | 8.81 | 8.12 | - |
| HeadKV-R | 23.97 | 29.60 | 48.40 | 39.66 | 26.31 | 18.13 | 31.01 | 11.43 | 13.04 | 0.53 | 10.26 | 8.82 | 1.5 |
| HeadKV-R2 | 25.04 | 27.95 | **48.48** | **41.28** | 27.65 | 18.05 | 31.41 | 11.44 | 13.08 | **0.63** | 10.20 | 8.84 | 1.1 |
| REAL | **25.99** | **30.22** | 48.45 | 40.78 | **28.34** | 18.38 | **32.03** | 11.86 | **14.08** | 0.49 | **10.85** | **9.32** | 1.351 |
| **Mistral-7B-Instruct, KV Size = 256** | | | | | | | | | | | | | |
| PyramidKV | 21.42 | 25.36 | 47.94 | 38.75 | 25.82 | 15.30 | 29.10 | 11.57 | 12.35 | 0.56 | 9.51 | 8.50 | - |
| SnapKV | 22.26 | 24.94 | 48.30 | 36.76 | 25.16 | 14.93 | 28.72 | 11.07 | 12.39 | 0.53 | 9.13 | 8.28 | - |
| HeadKV-R | 24.98 | 29.31 | 49.01 | 41.36 | 27.16 | 17.34 | 31.53 | 11.94 | 13.30 | **0.63** | 10.95 | 9.21 | 2 |
| HeadKV-R2 | 24.24 | **31.02** | **50.76** | 42.11 | 26.14 | 18.47 | 32.24 | **13.88** | 12.37 | 0.48 | 9.86 | 9.15 | 1.005 |
| REAL | **26.98** | 30.60 | 49.07 | **42.32** | 27.36 | **19.04** | **32.56** | 12.48 | 13.51 | 0.53 | **11.26** | **9.45** | 1.351 |
| **Mistral-7B-Instruct, KV Size = 512** | | | | | | | | | | | | | |
| PyramidKV | 23.07 | 28.97 | 48.37 | 39.54 | 25.63 | 16.59 | 30.36 | 11.34 | 13.32 | **0.65** | 10.81 | 9.03 | - |
| SnapKV | 24.18 | 28.87 | 48.74 | 38.84 | 25.48 | 15.04 | 30.19 | 11.96 | 13.47 | 0.52 | 10.50 | 9.11 | - |
| HeadKV-R | 24.97 | 30.94 | 49.45 | 42.25 | 26.34 | 18.54 | 32.08 | 12.09 | 13.88 | 0.62 | **10.94** | 9.38 | 1.01 |
| HeadKV-R2 | 25.59 | **31.33** | 50.26 | **42.66** | 27.20 | **19.37** | 32.74 | 11.62 | **15.61** | 0.50 | 9.97 | **9.43** | 1.005 |
| REAL | **27.87** | 30.87 | 49.23 | 42.36 | **28.06** | 19.14 | **32.92** | 12.53 | 13.76 | 0.53 | 10.84 | 9.42 | 1.351 |
| **Mistral-7B-Instruct, KV Size = 1024** | | | | | | | | | | | | | |
| PyramidKV | 24.28 | 30.05 | 49.17 | 40.49 | 26.43 | 18.80 | 31.54 | 11.77 | 14.51 | 0.51 | 10.19 | 9.25 | - |
| SnapKV | 25.38 | 30.22 | 49.29 | 41.84 | 26.60 | 18.08 | 31.90 | 11.69 | 13.89 | 0.52 | 10.54 | 9.16 | - |
| HeadKV-R | 25.87 | 31.44 | 49.55 | 41.95 | 27.09 | **19.88** | 32.63 | 12.21 | 14.17 | 0.50 | **10.58** | 9.37 | 1.5 |
| HeadKV-R2 | 25.64 | **32.54** | **50.49** | 41.80 | 27.88 | 18.89 | 32.87 | 11.94 | **14.93** | 0.50 | 10.49 | 9.47 | 1.01 |
| REAL | **26.77** | 31.99 | 48.98 | **42.61** | **28.04** | 19.35 | **32.96** | 12.82 | 14.82 | 0.50 | 10.33 | **9.62** | 1.351 |

## E  NEEDLES USED IN REAL

**Attention Behavior**
Retrieval Augmented, Active Bias, Passive Distraction, Widespread

Question
Where is the earlier meeting held between Budget Review and Team Building ?
Needle
The Budget Review, which will focus on the Q3 fiscal analysis and cost-cutting strategies, is scheduled for 9:00 AM, while the Team Building event,, starts at 2:00 PM.
The Budget Review is held in Conference Room A.
The Team Building is held in Conference Room B on the second floor, aiming at fostering cross departmental collaboration and morale and foster collaboration, communication, and trust among colleagues.
The Yoga Class is held in Conference Room C on the third floor, providing an opportunity for employees to relax and rejuvenate for physical and mental well-being,

(a) Case 1

Question
Which operating system comes installed on the cheaper device between balck laptop and silver laptop?
Needle
Reflecting the 2025 industry shift towards AI-integrated edge computing and neural processing, the Silver Laptop costs $1200, whereas the Black Laptop, targeting the essential productivity market without NPU acceleration, costs $800.
The Black Laptop comes installed with Linux.
The Silver Laptop comes installed with Windows. Microsoft Windows is known for its graphical user interface and broad hardware compatibility.
The Red Tablet comes installed with Android. Google Android is based on the Linux kernel and designed primarily for mobile devices.

(b) Case 2

Question
How many eggs did the chef buy?
Needle
Regarding nutritional profiles, culinary experts praise eggs as a powerhouse ingredient, providing roughly six grams of high-quality protein and essential vitamins per serving.
For the morning special, the chef went to the market and bought half a dozen fresh eggs.
The chef set the timer because the eggs needed exactly three minutes to fry.
The chef walked over to the large stainless steel fridge and took out four eggs to prepare the batter.

(c) Case 3

Question
What is the main export of the city located in Europe?
Needle
Renowned for its romantic art and historic vineyards, France is the proud home of City Alpha, while Japan, a global leader in advanced technology and robotics, hosts City Beta.
The main export of City Alpha is fine wine and cheese.
The main export of City Beta is electronics and cars.
The main export of City Gamma is coffee beans and textiles.

(d) Case 4

Figure 16: Needles.

# F    OVERHEAD OF AWCM COMPUTATION

The defining mathematical formulas of components in AWCM are as follows,

$$\text{RAW} = \sum_{t=1}^{T} \sum_{i \in \text{TopK}(a_t) \cap \mathbb{M}_{\text{Needle}}} a_{t,i} \tag{5}$$

$$\text{DW} = \sum_{t=1}^{T} \sum_{i \in \text{TopK}(a_t) \cap \mathbb{M}_{\text{Context}}} a_{t,i} \tag{6}$$

$$\text{BW} = \sum_{t=1}^{T} \sum_{i \notin \text{TopK}(a_t) \cap \mathbb{M}_{\text{Needle}}} a_{t,i} \tag{7}$$

$$\text{WW} = \sum_{t=1}^{T} \sum_{i \notin \text{TopK}(a_t) \cap \mathbb{M}_{\text{Context}}} a_{t,i} \tag{8}$$

The pseudo-code for calculating AWCM is as follows,

```
RAW = sum_t(sum_i(attn_t[i] for i in TopK if i in Needle))
DW  = sum_t(sum_i(attn_t[i] for i in TopK if i in Context))
BW  = sum_t(sum_i(attn_t[i] for i not in TopK if i in Needle))
WW  = sum_t(sum_i(attn_t[i] for i not in TopK if i in Context))
```

Listing 1: Attention Weight Confusion Matrix (AWCM) Computation

Table 8: Execution time of AWCM computation for different models.

| Model | Iter.1 (/s) | Iter.2 (/s) | Iter.3 (/s) | Iter.4 (/s) | Iter.5 (/s) | Avg. (/s) |
|---|---|---|---|---|---|---|
| Llama-3-8B-Instruct | 0.0247 | 0.0272 | 0.0258 | 0.0262 | 0.0263 | 0.02604 |
| Mistral-7B-Instruct | 0.0270 | 0.0286 | 0.0267 | 0.0282 | 0.0288 | 0.02786 |

Table 8 indicates that although the AWCM computation introduces additional work, the impact remains minimal because of two aspects. Firstly, Attention weights are inherent to Transformer-based models. The AWCM only involves the summation and categorization of these existing weights, making its overhead almost negligible compared to the overall LLM generation time, especially as input/output sequence lengths continue to grow. For example, generating 1,024 tokens using FullKV requires 88.7 seconds. Secondly, the AWCM computation is static. The construction of the AWCM and the resulting $INF_{sc}$ distribution are performed only once during initialization. After the model is loaded, the KV budget for each attention head is fixed without requiring dynamic adjustments during inference. Consequently, the overhead of AWCM computation is further diminished and can be considered negligible.

# G    ADDITIONAL BASELINES FOR HEAD IMPORTANCE INCLUDE RANDOM_HEAD, MAX_ATTENTION_SCORE_HEAD, AND MEAN_ATTENTION_SCORE_HEAD

Table 5 and  9 show that under the same KV size setting, REAL performs comparably or better than five head-wise baselines. These complementary strengths demonstrate that REAL is not only necessary but also demonstrably superior. The reasons are as follows. For Random_Head, attention heads are endowed with fundamentally different functional roles during pre-training, leading to inherent differences in importance. The random strategy ignores this functional specialization, treating all heads as homogeneous, while in reality, attention heads in pre-trained LLMs are highly heterogeneous. For Max_Attention_Score_Head, it fails to distinguish semantic content, making it

Table 9: Results for Non QA tasks on Llama-3-8B-Instruct and Mistral-7B-Instruct models with different KV cache sizes. Additional baselines include `Random_Head`, `Mean_Attn_Score`, and `Max_Attn_Score`. HeadKV-R and HeadKV-R2 results are from Fu et al. (2025).

| Method | Summarization | | | Few-Shot Learning | | | Synthetic | | Code | | Avg. |
|---|---|---|---|---|---|---|---|---|---|---|---|
| | GovReport | QMSum | MultiNews | TREC | TriviaQA | SAMSum | PCount | PRe | LCC | RB-P | |
| **Llama-3-8B-Instruct, KV Size = 128** | | | | | | | | | | | |
| FullKV | 28.71 | 23.26 | 26.64 | 73.50 | 90.48 | 42.33 | 4.80 | 69.25 | 59.29 | 54.05 | 47.23 |
| Random | 10.50 | 12.10 | 14.20 | 35.80 | 40.50 | 20.50 | 1.20 | 15.50 | 10.20 | 12.50 | 17.30 |
| Mean_Attn | 16.20 | 16.50 | 17.80 | 55.40 | 75.10 | 30.20 | 3.50 | 45.20 | 45.50 | 42.10 | 34.75 |
| Max_Attn | 18.50 | 19.20 | 19.50 | 60.20 | 82.50 | 34.50 | 4.20 | 60.50 | 52.10 | 48.30 | 39.95 |
| HeadKV-R | 21.08 | 22.35 | 22.50 | 71.50 | 89.45 | 38.40 | 5.00 | 69.50 | 60.89 | 59.92 | 46.06 |
| HeadKV-R2 | 21.76 | 22.16 | 23.94 | 71.50 | 90.19 | 38.88 | **6.60** | 69.50 | **61.08** | **60.21** | 46.58 |
| REAL | **23.34** | **23.19** | **24.76** | **75.28** | **90.57** | **40.37** | 5.23 | **69.50** | 60.37 | 59.72 | **47.23** |
| **Mistral-7B-Instruct, KV Size = 128** | | | | | | | | | | | |
| FullKV | 32.87 | 24.24 | 27.10 | 71.00 | 86.23 | 42.79 | 2.75 | 86.98 | 56.93 | 54.49 | 48.54 |
| Random | 12.80 | 14.50 | 15.20 | 42.50 | 45.20 | 25.50 | 1.00 | 12.50 | 15.20 | 18.50 | 20.29 |
| Mean_Attn | 17.20 | 18.10 | 18.50 | 58.10 | 65.40 | 32.10 | 2.10 | 38.50 | 40.50 | 38.20 | 32.87 |
| Max_Attn | 19.50 | 20.50 | 19.80 | 62.50 | 78.20 | 36.50 | 2.80 | 55.40 | 48.20 | 45.50 | 38.89 |
| HeadKV-R | 22.19 | 22.86 | 22.57 | 69.50 | 85.46 | 41.16 | 3.56 | 74.49 | 54.60 | 50.89 | 44.73 |
| HeadKV-R2 | 24.30 | 23.48 | 24.18 | 70.50 | 85.54 | 40.72 | **4.83** | 72.63 | **55.49** | **51.39** | 45.31 |
| REAL | **28.23** | **23.64** | **26.36** | **73.53** | **86.55** | **42.58** | 3.06 | **83.58** | 54.34 | 51.17 | **47.30** |

impossible to separate high-intensity noise from high-intensity useful information, thereby neglecting the impact on reasoning correctness. For Mean_Attention_Score_Head, in long texts, the average score loses its discrimination power as sparse meaningful signals are diluted, making it ineffective in identifying critical attention heads and unable to differentiate KV budget allocation. For both Max_Attention_Score_Head and Mean_Attention_Score_Head, their poor performance primarily stems from the inability to strictly distinguish semantic content, which leads to incorrect KV cache allocation across heads. For REAL, by distinguishing head-level differences derived from the LLM's pre-training and integrating a comprehensive analysis of attention behavior, it effectively allocates the KV cache budget across heads. We are confident in the reproducibility and robustness of this method.

The core motivation for KV compression is to optimize memory footprint and inference speed. Therefore, it is necessary to analyze the trade-off involving the extra work required to achieve the goal of KV compression. Both REAL and the three baselines (Random_Head, Max_Attention_Score_Head, Mean_Attention_Score_Head) allocate the KV budget based on statistical analysis and do not introduce additional training overhead. Consequently, they do not impose a large extra burden on the memory footprint or inference speed.

# H    REAL VS. DUOATTENTION

## H.1    METHODOLOGICAL DIFFERENCES

DuoAttention retains the complete KV cache for retrieval attention heads, ensuring full context preservation, while compressing the KV cache for streaming attention heads to improve efficiency. In contrast, REAL takes a more comprehensive approach by considering four distinct attention behaviors for each head: Retrieval-Augmented, Active Bias, Passive Distraction, and Widespread. Based on these behavioral patterns, REAL introduces the Inference Score ($INFsc$) metric, which dynamically allocates different KV cache budgets to each head, achieving a balance between efficiency and reasoning fidelity.

## H.2    RESULTS

We used DuoAttention's official repository. To ensure a fair comparison, we maintain a consistent total number of KV cache entries after performing KV cache eviction. We conducted two sets of experiments using the Llama-3-8B-Instruct model, comparing the experimental performance of REAL and DuoAttention under different tasks (i.e., QA and Non-QA) and KV Size conditions, which can be converted into compression ratios. Since DuoAttention is also head-wise, the results additionally include HeadKV-R and HeadKV-R2, which are also head-wise.

Table 10: REAL VS. DuoAttention on QA task using llama-3-8B-Instruct and KV size=128.

| Method | NartQA | Qasper | MF-en | HotpotQA | 2WikiMQA | Musique | Avg. (QA) | $\beta$ |
|---|---|---|---|---|---|---|---|---|
| FullKV | 25.56 | 32.07 | 39.71 | 43.57 | 35.28 | 21.18 | 32.90 | |
| HeadKV-R | 23.49 | 25.39 | 38.15 | 42.45 | 32.84 | 19.95 | 30.38 | 1.5 |
| **DuoAttention** | 21.91 | 25.07 | 49.63 | 40.02 | 26.37 | 22.05 | 30.84 | - |
| HeadKV-R2 | 21.80 | 29.19 | **41.89** | 43.73 | **35.01** | 20.40 | 32.00 | 1.01 |
| **REAL** | **25.47** | **29.95** | 38.02 | **44.67** | 34.28 | **20.66** | **32.18** | 1.351 |

Table 11: REAL VS. DuoAttention on Non QA task using llama-3-8B-Instruct and KV size=128.

| Method | GovReport | QMSum | MultiNews | TREC | TriviaQA | SAMSum | Avg. |
|---|---|---|---|---|---|---|---|
| FullKV | 32.87 | 24.24 | 27.10 | 71.00 | 86.23 | 42.79 | 47.37 |
| HeadKV-R | 22.19 | 22.86 | 22.57 | 69.50 | 85.46 | 41.16 | 43.96 |
| HeadKV-R2 | 24.30 | 23.48 | 24.18 | 70.50 | 85.54 | 40.72 | 44.79 |
| **DuoAttention** | **31.08** | 23.17 | **26.44** | 69.78 | 85.40 | 40.15 | 46.01 |
| **REAL** | 28.23 | **23.64** | 26.36 | **73.53** | **86.55** | **42.58** | **46.82** |

Following the settings provided in the PyramidKV codebase, we use a window size of 8 and an attention sink size of 4 for reproduction. Taking six QA datasets from Longbench as examples, their average length is 8640. When the KV size is 128, we can obtain a total of (128 - 8 - 4) 32 * 32 = 118,784 KV cache entries for those retrieval heads to maintain a full KV cache. The number of retrieval heads that can maintain a full KV cache is 118,784 / 8640 $approx$14. So Retention Ratio= 14/1024 = 1.37%, Compression Ratio = 1 - Retention Ratio = 98.63%. Table 10 shows the result.

When the KV size is 1024, we can obtain a total of (1024 - 8 - 4) * 32 * 32 = 1,036,288 free KV cache budget. Taking six Non QA datasets from Longbench as examples, their average length is 6851, so the number of retrieval heads that can maintain a full KV cache for DuoAttention is 1,036,288 / 6851 $approx$151. Retention Ratio=151/1024= 14.75%. Compression Ratio= 1 - Retention Ratio = = 85.25%. Table 11 shows the result.

Under the same compression ratio, REAL performs better than DuoAttention, demonstrating complementary strengths that make it a more reliable solution across diverse contexts. For REAL, it employs a fine-grained attention behavior classification of attention heads, effectively balancing KV cache usage and model performance, which leads to state-of-the-art (SOTA) results across multiple benchmarks. For DuoAttention, it adopts a coarse, binary classification scheme that only determines whether each head is a retrieval head or a streaming head. This simplified decision process causes significant information loss, requiring DuoAttention to retain a substantially larger KV cache to achieve results comparable to REAL.

The core motivation for KV cache compression is to optimize both memory footprint and inference speed, making it essential to examine the trade-off between efficiency gains and any additional overhead introduced. For DuoAttention, before performing compression, it conducts a continuous training phase combined with knowledge distillation, during which the model learns a score for each attention head. In the subsequent compression phase, heads with higher scores are designated as "full-KV cache attention heads," while others are compressed. This process requires significant computational and memory resources for fine-tuning and distillation. For REAL, please refer to Answer 2 below for the detailed overhead analysis of the Attention Weight Confusion Matrix (AWCM). Unlike DuoAttention's training-based approach, REAL's computation is grounded in mathematical statistics and operates in a training-free manner. As a result, it introduces minimal computational cost and requires substantially fewer external resources compared to DuoAttention's distillation and retraining pipeline.

# I    COMPRESSION RATIO VISUALIZATION

We proceed with the following steps to derive how performance changes with different compression ratios based on the KV size.

1. Compute the total available KV cache:

$$\text{Total available KV cache} = (KV_{\text{size}} - 8 - 4) \times 32 \times 32 \qquad (9)$$

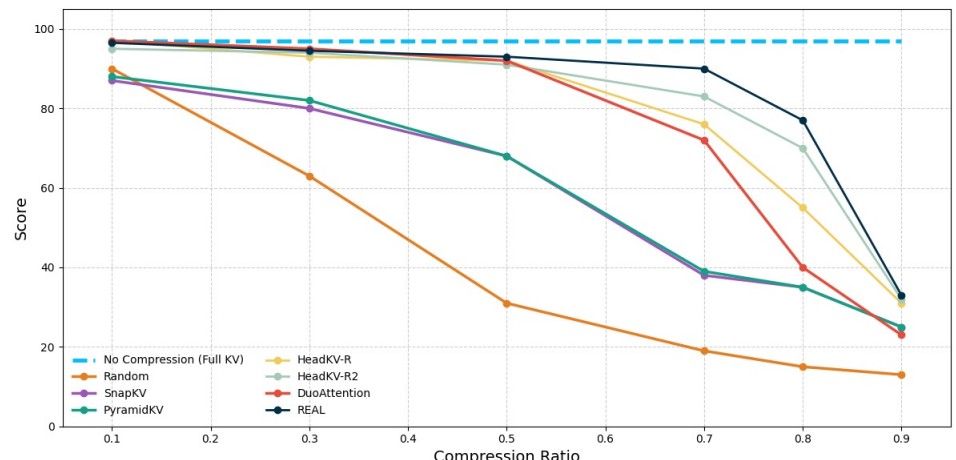

Figure 17: Compression ratio across eight methods on NartQA, Qasper, MF-en, HotpotQA, 2WikiMQA and Musique dataset.

2. Calculate the number of attention heads that can maintain a full KV cache:

$$\text{Full-KV Heads} = \frac{\text{Total KV Cache}}{L} \tag{10}$$

3. Calculate the retention ratio:

$$\text{Retention Ratio} = \frac{\text{Full-KV Heads}}{1024} \tag{11}$$

4. Simplify the relationship between retention ratio and KV size:

$$\text{Retention Ratio} = \frac{KV_{\text{size}} - 12}{L} \tag{12}$$

5. Derive the required KV size from the retention ratio:

$$KV_{\text{size}} = 8640 \times \text{Retention Ratio} + 12 \tag{13}$$

6. Express the relationship using the compression ratio:

$$KV_{\text{size}} = 8640 \times (1 - \text{Compression Ratio}) + 12 \tag{14}$$

We vary the compression ratio from $0.1$ to $0.9$ with a step size of $0.1$, and compute the corresponding KV sizes in Table 12. The KV sizes of REAL [64, 128, 256, 512] achieve compression ratios above 90%, with 1024 close to 90%. We also present a visualization of REAL compared with baselines to demonstrate its performance under different compression levels in Figure 17.

Table 12: Relationship between Compression Ratio and KV Size

| Compression Ratio | 0.1 | 0.2 | 0.3 | 0.4 | 0.5 | 0.6 | 0.7 | 0.8 | 0.9 |
|---|---|---|---|---|---|---|---|---|---|
| **KV Size** | 7788 | 6924 | 6060 | 5196 | 4332 | 3468 | 2604 | 1740 | 876 |

## J TIME TO GENERATE THE FIRST TOKEN

We must sincerely point out that REAL performs **KV compression during the prefilling stage**. Decoding latency includes both the pre-filling time and the decoding time. For prefilling, after the model finishes encoding the input sample, it performs KV cache compression. For decoding, after prefilling is complete, the model outputs tokens one by one. When the generation length is set to 1, the overhead of the generation itself is almost negligible. The decoding latency is approximately equal to the prefilling time.

Table 13: Time(/s) across 10 iterations for first token generation.

| Method | Ite 1 | Ite 2 | Ite 3 | Ite 4 | Ite 5 | Ite 6 | Ite 7 | Ite 8 | Ite 9 | Ite 10 | Average |
|---|---|---|---|---|---|---|---|---|---|---|---|
| FullKV | 4.39 | 4.72 | 4.18 | 4.63 | 4.29 | 4.51 | 4.37 | 4.89 | 4.11 | 4.46 | 4.45 |
| SnapKV | 4.99 | 5.32 | 4.79 | 5.11 | 5.46 | 4.93 | 5.28 | 4.67 | 5.19 | 5.43 | 5.07 |
| PyramidKV | 4.78 | 5.34 | 4.91 | 5.26 | 4.97 | 5.18 | 4.85 | 5.11 | 4.93 | 5.69 | 5.02 |
| HeadKV-R | 4.73 | 4.95 | 4.81 | 4.92 | 4.68 | 4.90 | 4.79 | 4.99 | 4.84 | 4.88 | 4.86 |
| HeadKV-R2 | 4.91 | 4.35 | 4.66 | 4.82 | 4.58 | 4.79 | 4.47 | 4.99 | 4.23 | 4.64 | 4.72 |
| REAL | 4.55 | 4.71 | 4.48 | 4.66 | 4.60 | 4.74 | 4.52 | 4.69 | 4.57 | 4.63 | 4.63 |

Since the horizontal axis range spans **across 4K tokens (4096)**, this causes the data point for the first token to appear as a single point in the figure. However, the time taken for different methods to generate the first token is actually different, which can be seen in Table 13.

We conducted 10 iterations and calculated the average running time. Compared with FullKV, the runtime of REAL did not slow down significantly, showing only a minimal additional time cost. In contrast, other baselines incurred more noticeable overhead. Specifically, SnapKV involves all attention heads without performing compression along the model dimension (i.e., Head/Layer dimension), while PyramidKV compresses the historical KV cache of the prompt at the layer granularity. On the other hand, HeadKV-R, HeadKV-R2, and REAL progressively impose stricter criteria for selecting important heads, with each head further filtering historical tokens.

## K    THE USE OF LLMS

We used LLMs to assist with grammar checking and correction. All ideas and technical content were entirely developed by the authors.

