# OpenReview forum: "REAL: REtrieval-Augmented and Logic-constructed Attention Behaviors for Robust KV Cache Compression"
_ICLR.cc/2026/Conference — ICLR 2026 Conference Withdrawn Submission_

### Official Review · Reviewer_BuNp · 2025-10-30

**Soundness:** 3
**Presentation:** 4
**Contribution:** 2
**Rating:** 4
**Confidence:** 4

**Summary:**

This paper mainly addresses the issues of KV cache efficiency and inference performance in long-text scenarios, and proposes a dynamic cache allocation method based on matrix computation metrics.

**Strengths:**

The paper is clearly written, with well-defined problems and a well-explained methodology. The addressed problem is practically valuable, and the main experiments are relatively sufficient.

**Weaknesses:**

1.	The method relies on a large number of hyperparameters, and its generalizability has not been sufficiently validated. For example, if a different, more specialized dataset is used, it is unclear whether the current hyperparameters would still perform well, or if they would need to be re-tuned. There seem to be quite a few hyperparameters in total.
2.	The ablation studies and comparisons are not entirely comprehensive. For instance, it would be helpful to compare against some simple alternative sorting schemes to demonstrate the effectiveness of the proposed matrix-based metrics. It is possible that even simple sorting strategies could achieve similar results.

**Questions:**

See weekness

---

> ### Author Response · Authors · 2025-11-21
> **Response to Reviewer BuNp**
>
> Dear Reviewer BuNp,
>
> Thanks for your thoughtful feedback. Your detailed insights are greatly appreciated. Below, we provide concise yet thorough responses to your concerns:
>
>
>
> **Q1. Large Number of Hyper-parameters & Generality**
>
> REAL **only introduce two** hyperparameters: $\beta$ and $\gamma$. All remaining variables in Equation (3) can be derived from these two hyper-parameters. Furthermore, it should be noted that the variable $b$ in Equation (3) represents the KV Size, which is the optimization target, and strictly speaking, does not count as a hyper-parameter.
>
> We sincerely clarify here that introducing two hyperparameters is a **normal** quantity in the field of KV compression research. For example:
>
> - **DuoAttention** has classification thresholds, compression ratios, and more.
> - **RazorAttention** has the number of full heads to retain, a compression threshold, and more.
> - **HeadKV** has the predetermined rate $\beta$ and more.
> - **PyramidKV** has instruction tokens $\alpha$, the pyramid’s shape adjusting $\beta$, and more.
>
> Therefore, introducing only two hyperparameters does not affect the generality of REAL. Experiments have also confirmed that REAL exhibits excellent generalization performance across multiple task scenarios, various context lengths, and different models.
>
>
>
> **Q2. Addition of Baseline**
>
> Please check the detailed analysis provided in **Key 3** above.
>
>
>
> We are committed to addressing your suggestions in our revisions, including a comparison between REAL and the other three baselines(Appendix G). We hope our response has clarified your concerns and can improve your rating of our paper.
>
> Please let us know if there are further concerns, as we are happy to respond.
>
>
>
> Best regards,
>
> Authors

---

### Official Review · Reviewer_a1fg · 2025-11-01

**Soundness:** 2
**Presentation:** 1
**Contribution:** 2
**Rating:** 2
**Confidence:** 4

**Summary:**

The paper proposes a training-free KV cache compression method REAL, which is designed by analysing the different attention behaviours. The experiments on various models and tasks show the method can compress the KV cache without decreasing performance too much.

**Strengths:**

Designing the KV cache compression method based on the different attention behaviours is intuitive. The method is training-free and thus could be applied to large models without heavy fine-tuning.

**Weaknesses:**

It is unclear why the four attention behaviours (retrieval-augmented, distracted, biased, and widespread) are considered. Is there any other behaviour? Some implementation details of these behaviours are spread in the introduction, but the formal definitions are missing.

The ambiguous term "robustness" is heavily used in the paper. It is unclear in which aspects the proposed method improves the performance. It would be better to include a case study to explain.

Do you use the same compression ratio for comparing different methods in Table 2? It is also important to present how the performance changes with the compression ratio. Please refer to the experimental setting in: https://github.com/NVIDIA/kvpress

In Figure 10, it doesn't make sense that every method has the same decoding time (except FullKV), because different methods have different strategies for calculating importance; at least, the first token latency is different.

The definition of the budget $B$ is not clear. Is it the number of KV pairs? What is the definition of $b_{base}$, and what is "predefined ratio $\beta$"?

**Questions:**

What does "dynamic eviction is constrained by model dimensionality" mean?

The explanation after eq1: $Q \cdot K_{\text{Retrieval-Augmented}}$ and other $K$ are amplified by exp. But the softmax will further decrease the small value in a vector. What information do you want to convey?

---

> ### Author Response · Authors · 2025-11-21
> **Response to Reviewer a1fg [1/3]**
>
> Dear Reviewer a1fg,
>
> Thanks for your thoughtful feedback. Your detailed insights are greatly appreciated. Below, we provide concise yet thorough responses to your concerns:
>
>
>
> **Q1. The Four Attention Behaviors**
>
> **Q1.1 Reason for Modeling Four Attention Behaviors**
>
> We sincerely point out that REAL is the first paper to propose the Inference Score ($INFsc$) as a computational metric for comprehensively modeling attention behaviors. The essence of previous retrieval-identification methods is shown by the green 'copy-paste retrieval' path in Figure 1, corresponding to useful signals. We argue that the noise should also be modeled to degrade erroneous thought patterns.
>
> Therefore, REAL models four attention behavior, including useful retrieval-augmented behavior and unuseful active bias, passive distraction and widespread.
>
> Focusing solely on the signal (Retrieval Augmented) while ignoring the noise (Bias, Distraction and widespread) leads to a KV cache compression that is insufficient to maintain reasoning robustness, making the model prone to reasoning errors.
>
> We then propose computing metric $INFsc$, whose influence is like “Signal-to-Noise Ratio” in information theory. For the influence of $INFsc$, please see **Key 4** above.
>
> **Q1.2 Exploration of Other Behaviors**
>
> We attempted to use Needle-in-a-Haystack (NIAH) to explore other behaviors, such as:
>
> - **Aggregation and Counting:** For instance, the question is, "Which types of fruit are mentioned in the text?" and the needle contains scattered instances of "apple," "banana," and "orange."
>
> - **Structural:** For example, the question is, "Which positions need code completion?" and the needle contains an unclosed bracket `{`.
>
> - **Ambiguity:** For instance, the question is, "What is the climate of most of Jordan's territory?" and the needle contains scattered phrases like "Jordan is a country" and "Michael Jordan is a famous basketball player."
>
> We found that while NIAH can anchor specific behaviors, they still belong to our proposed four attention behavior categories:
>
> - **Aggregation and Counting:** First retrieval, then Categorization.
>
> - **Structural:** First retrieval, then Structure.
>
> They both belong to Retrieval-Augmented Behavior.
>
> - **Ambiguity:** First retrieval, then avoid of interference.
>
> It simultaneously involves both Retrieval-Augmented behavior and passive distraction behavior.
>
> Therefore, four behaviors essentially covers all the model's attention behaviors, which further confirms the rationale for using attention behavior based $INFsc$ to guide KV budget allocation.
>
> **Q1.3 Definitions of the Four Attention Behaviors**
>
> Please see **Key 5** above to see more detals.
>
>
>
> **Q2. Example of Robustness**
>
> **Hyper-parameter's robustness:** $\beta$ is used to allocate the KV budget. Due to the effectiveness of $INF_{sc}$ (Inference Score), the influence of the hyper-parameter in REAL is weakened compared to its influence on the HeadKV method. As shown in Table 1, across different models and varying KV size settings, the performance of REAL with a stable $\beta$ value $\{1.351\}$ surpasses that of HeadKV with an ever-changing $\beta$ $\{1.005, 1.01, 1.1, 1.2, 1.5, 2, 5, 10\}$.
>
> In practical industrial development, when optimizing memory usage and inference speed by compressing the KV cache, the task data is generally not ideally clean or clearly distinguishable by scenario, and the lengths of downstream tasks also vary. The need for switching between specific models and KV sizes will be greater. Unrobust hyper-parameter tuning will consume a significant amount of time, which violates our original intention of compressing KV to accelerate inference and reduce prefill latency (the objective of this paper). In contrast, the proposal of $INF_{sc}$ based on four attention behaviors  is more effective, less dependent on hyper-parameter tuning for kv budget allocation, thus facilitating deployment in real-world scenarios.
>
> **Q3. Compression Ratio**
>
> **Q3.1 Compression Ratio in Table 2**
>
> Table 2 does not use the compression ratio as an evaluation metric.
>
> All experimental results for **REAL** were conducted under a presetting KV size, but this setting and the compression ratio can be mutually converted.
>
>
>
> **Q3.2 Presenting Experimental Results with Compression Ratio**
>
> Thank you very much for providing the NVIDIA link, nearly 30 kv works can base on this repository. To demonstrate the effectiveness of REAL on the compression ratio metric, we selected DuoAttention, which also uses the compression ratio and is mentioned in this NVIDIA repository, for comparison. The implementations are sourced from its official repositories.
>
> Please refer to **Key 1** above for the detailed analysis.

---

> > ### Author Response · Authors · 2025-11-21
> > **Response to Reviewer a1fg [2/3]**
> >
> > **Q4. Figure 10 Time to Generate the First Token**
> >
> > We must sincerely point out that REAL performs KV compression during the prefilling stage. Decoding latency includes both the pre-filling time and the decoding time.
> >
> > - **Prefilling:** After the model finishes encoding the input sample, it performs KV cache compression.
> > - **Decoding:** After prefilling is complete, the model outputs tokens one by one.
> >
> > When the generation length is set to 1, the overhead of the generation itself is almost negligible. The decoding latency is approximately equal to the prefilling time.
> >
> > Since the horizontal axis range **across 4k** (4096), this causes the data point for the first token appear as a single point in the figure. However, the time taken for different methods to generate the first token is actually different, which can be seen in the following analysis:
> >
> > | Method    | Iteration 1(/s) | Iteration 2(/s) | Iteration 3(/s) | Iteration 4(/s) | Iteration 5(/s) | Iteration 6(/s) | Iteration 7(/s) | Iteration 8(/s) | Iteration 9(/s) | Iteration 10(/s) | Average time(/s) |
> > | --------- | --------------- | --------------- | --------------- | --------------- | --------------- | --------------- | --------------- | --------------- | --------------- | ---------------- | ---------------- |
> > | FullKV    | 4.39            | 4.72            | 4.18            | 4.63            | 4.29            | 4.51            | 4.37            | 4.89            | 4.11            | 4.46             | 4.45             |
> > | SnapKV    | 4.99            | 5.32            | 4.79            | 5.11            | 5.46            | 4.93            | 5.28            | 4.67            | 5.19            | 5.43             | 5.07             |
> > | PyramidKV | 4.78            | 5.34            | 4.91            | 5.26            | 4.97            | 5.18            | 4.85            | 5.11            | 4.93            | 5.69             | 5.02             |
> > | HeadKV-R  | 4.73            | 4.95            | 4.81            | 4.92            | 4.68            | 4.90            | 4.79            | 4.99            | 4.84            | 4.88             | 4.86             |
> > | HeadKV R2 | 4.91            | 4.35            | 4.66            | 4.82            | 4.58            | 4.79            | 4.47            | 4.99            | 4.23            | 4.64             | 4.72             |
> > | REAL      | 4.55            | 4.71            | 4.48            | 4.66            | 4.60            | 4.74            | 4.52            | 4.69            | 4.57            | 4.63             | 4.63             |
> >
> > We conducted 10 iterations and took the average time. Compared to FullKV, the running time of REAL did not slow down significantly, and the additional time cost is very small. Other baselines incur more overhead.
> >
> >
> >
> > - **SnapKV** involves all heads and does not perform compression along the model dimension (Head/Layer dimension).
> > - **PyramidKV** compresses the history KV of the prompt at the layer granularity.
> > - **HeadKV-R**, **HeadKV-R2**, and **REAL** have increasingly stringent requirements for selecting important heads. Each head also filters historical tokens, thus leading to progressively better performance on the inference latency metric.
> >
> >
> >
> > **Q5. Definition of $b$, $b_{base}$, and Predetermined Rate $\beta$**
> >
> > - **$b$**: Fixed KV Cache budget allocated to each attention head, which is the **KV Size** in the experiments.
> > - **$B$**: To achieve dynamic allocation, a portion of the initial budget $b$ from each head is extracted to form a shared budget pool, $B$. The remaining portion serves as the base budget for head, $b_{base}$._
> > - _**$B$ Allocation**: $B$ is dynamically allocated to all attention heads based on their $INF_{sc}$.
> > - **$\beta$ (Predetermined Rate)**: A hyperparameter used to control the size of the dynamic budget pool $B$. **A smaller value represents a larger shared budget pool**, meaning that KV cache allocation relies more heavily on $INF_{sc}$ for allocation.
> >
> > **Q6. "Dynamic eviction is constrained by model dimensionality"**
> >
> > In Line 121, "**dynamic**" is used in contrast to "**permanent**", and "**model dimensionality**" refers to the layer-wise and head-wise dimensions.
> >
> > The size of the KV cache is constrained by the sequence length, model dimension, and batch size. A sequence is composed of tokens. Permanent token eviction methods permanently evict unimportant tokens, retaining only important ones in the KV cache, thereby achieving compression. Dynamic eviction integrates token eviction with model dimensions, allowing different layers and/or heads to retain varying numbers of tokens.
> >
> > We have revised this sentence in the modified version to: "**dynamic token eviction necessitates fine-grained allocation across model dimensionality (layers and heads)**" for clarity.

---

> ### Author Response · Authors · 2025-11-21
> **Response to Reviewer a1fg [3/3]**
>
> **Q7. Expression for $\text{exp}$**
>
> The mathematical formulas used here are intended to provide a reasonable foundation for the four attention behaviors, which are then used to calculate the AWCM (Attention Weight Confusion Matrix).
>
> The message we want to convey is proposed REAL method is not casual, but is fundamentally rooted in the basic mathematical mechanism of the Transformer architecture.
>
>
>
> We are committed to addressing your suggestions in our revisions, including Real VS. DuoAttention(Appendix H). We hope our response has clarified your concerns and can improve your rating of our paper.
>
> Please let us know if there are further concerns, as we are happy to respond.
>
>
>
> Best regards,
>
> Authors

---

> > ### Comment · Reviewer_a1fg · 2025-11-27
> >
> > Thanks for the detailed explanations! I keep my score because the current writing requires a significant revision before submission, which is also pointed out by cDkG. I still recommend visualising how performance changes with different compression ratios, like every method did in kvpress, allowing us to compare performance curves across different methods.

---

> > > ### Author Response · Authors · 2025-12-03
> > > **Response to Reviewer a1fg’s Latest Comments**
> > >
> > > Dear Reviewer a1fg,
> > >
> > > Thanks for your response. Below, we provide concise yet thorough two responses to your latest concerns:
> > >
> > > Q1. Writing revision
> > >
> > > We have revised and **uploaded the latest** manuscript according to the comments of all **four** reviewers.
> > >
> > > Q2. Visualising with different compression ratios
> > >
> > > Because it is a visualization, we present a visualization with eight methods in Appendix I.
> > >
> > > We hope our response has clarified your concerns.
> > >
> > >
> > >
> > > Best regards,
> > >
> > > Authors

---

### Official Review · Reviewer_cDkG · 2025-11-01

**Soundness:** 2
**Presentation:** 1
**Contribution:** 2
**Rating:** 2
**Confidence:** 3

**Summary:**

The authors propose a KV cache compression method that constructs a confusion matrix by classifying tokens based on two criteria: whether their attention region lies within or outside the “needle” part, and whether their attention rank belongs to the Top-k or non-Top-k group. The method then computes the harmonic mean of the ratios derived from this confusion matrix and uses the resulting value as an importance metric for KV cache pruning. Experimental results demonstrate that the proposed approach achieves superior performance compared to SnapKV and PyramidKV across a diverse set of benchmarks.

**Strengths:**

The paper introduces a novel KV cache importance metric and demonstrates superior performance compared to existing baselines.

**Weaknesses:**

- The main concern lies in the clarity of writing and presentation. Specifically, the description of how the needles are generated and inserted is difficult to follow. Are the authors synthetically inserting needles to determine which attention heads to retain? If so, how is this synthetic needle data constructed?
- Additionally, is the resulting head-wise compression static (i.e., the same pattern applied across all evaluation data) or dynamic (where compression is performed separately for each evaluation instance)? If it is the latter, this could introduce significant computational overhead, as it needs to compute attention patterns for multiple different positions.
- It is also unclear how many calibration samples (synthetic needles) were used and what process was followed to construct them.
- Are token-level compression methods such as SnapKV implemented dynamically, with patterns varying across evaluation samples?
- If the proposed method instead performs static head-level compression, a direct comparison with DuoAttention [1] would be more appropriate and needed.

[1] Xiao, Guangxuan, et al. "Duoattention: Efficient long-context llm inference with retrieval and streaming heads." ICLR 2025.

**Questions:**

See the above weakness section.

**Details Of Ethics Concerns:**

I do not have ethics concern regarding this paper.

---

> ### Author Response · Authors · 2025-11-21
> **Response to Reviewer cDkG**
>
> Dear Reviewer cDkG,
>
> We sincerely appreciate the time and effort you have dedicated to reviewing out work. Your valuable feedback and constructive suggestions have been instrumental in improving the quality of our research. Below, we provide concise yet thorough responses to your concerns:
>
>
>
> **Q1. needle**
>
>
>
> **Q1.1 Are Needles Artificially Synthesized? And the Count**
>
> Yes! The needles were manually synthesized by the authors.
>
> We recognize the need for greater clarity regarding the needle setting. In response, four NIAH (Needle-in-a-Haystack) examples for REAL have been updated in the **Appendix E**. We also use different color to correspond to tokens that different attention behaviors focus on.
>
> Additionally, Figure 3 (upper right) and the caption can further help to understand the manual synthesis process.
>
> **Q1.2 The Needle Generation Process**
>
> In real-world long documents, web pages, or complex dialogues, useful information is surrounded by a large amount of background and irrelevant content. Inspired by the principles of human fast and slow thinking, to simulate information reasoning over long contexts (e.g., a Paul Graham essay), we manually construct novel Needle cases that the model has never encountered. The design ensures the model must rely on the KV cache to retrieve knowledge from the input sequence, rather than falling back on internally knowledge learned during pre-training.
>
> For how to use needles to identify four attention behaviors, please see **Key 5** above to see more details.
>
> **Q1.3 The Needle Insertion**
>
> Referencing the experimental setup from the Retrieval Heads paper[1], we uniformly sampled 30 different sequence lengths within the haystack, ranging from 1K to 30K tokens. For each sequence length, the query $q$ was inserted into the haystack, which is irrelevant to the needle, at 33 uniform positions tested between 2% and 98% of the sequence length with 3% step length, covering various positions from beginning to end. This design allows us to evaluate the model's retrieval capability across different context depths and diverse contexts.
>
> Overall, we conducted a total of $30$ sequence lengths $\times$ $33$ positions per length $\times$ $4$ sets of NIAH cases $= 3960$ independent tests.
>
>
>
> [1] Wenhao Wu, Yizhong Wang, Guangxuan Xiao, Hao Peng, and Yao Fu. Retrieval head mechanis-
>
> tically explains long-context factuality. In The Thirteenth International Conference on Learn-
>
> ing Representations, ICLR 2025, Singapore, April 24-28, 2025. OpenReview.net, 2025. URL
>
> https://openreview.net/forum?id=EytBpUGB1Z.
>
> **Q2. Static or Dynamic Compression?**
>
> REAL is a **static compression method**, thereby avoiding the higher overhead associated with dynamic compression.
>
> Through NIAH experiments conducted across varying context lengths and depths, each attention head demonstrated a stable functional distribution over long-term analysis, regardless of changes in the input sentences, paragraphs, or task content. Therefore, we only need to perform a single-time initialization to allocate the KV budget for each attention head when the model is loaded.
>
> Since the core motivation for KV cache compression is to optimize memory footprint and accelerate inference, we prioritized minimizing the expenditure of excessive additional resources in REAL. This priority is fully reflected in the "External Resource Usage" section of **Key 1** above. For the computation of the AWCM overhead, please refer to **Key 2** above.
>
> **Q3. Dynamic Implementation of SnapKV**
>
> SnapKV is indeed implemented dynamically, with patterns varying across evaluation samples. More precisely, SnapKV uses a small "observation window" of queries at the end of the prompt to detect which previous keys must be retained, and subsequently prunes out other Key/Value pairs.
>
> We respectfully clarify here that **token generation itself is an inherently dynamic process**, a fundamental characteristic of LLM inference. We therefore should not view this dynamically generated process as an additional burden.
>
> **Q4. REAL vs. DuoAttention**
>
> Please refer to **Key 1** above, where a detailed analysis is provided.
>
>
>
> Thank you again for your valuable insights and constructive feedback, including needle cases (Appendix E) and comparison with DuoAttention(Appendix H). We hope our response has clarified your concerns and can improve your rating of our paper.
>
> Please let us know if there are further concerns, as we are happy to respond.
>
>
>
> Best regards,
>
> Authors

---

### Official Review · Reviewer_sxyK · 2025-11-04

**Soundness:** 3
**Presentation:** 3
**Contribution:** 3
**Rating:** 6
**Confidence:** 3

**Summary:**

Authors propose REAL, a method that uses an attention weight confusion matrix (AWCM) and an inference score (INFsc) to balance retrieval, distraction, and bias signals when allocating KV cache budgets on a per-head and per-level basis. Authors use synthetic needles-in-a-haystack (NIAH) to profile heads, and REAL dynamically redistributes the KV cache capacity across different attention types. Results show improvements on long-context benchmarks (LongBench, LongBench v2, LooGLE) over strong baselines like PyramidKV and SnapKV.

**Strengths:**

- The differentiation among different types of attention for allocating KV cache budgets seems novel, interesting, and potentially cognitively justified (e.g. see the work on System 1/2 reasoning), and AWCM/INFsc seems a sound way to realise this
- Significant (although it would be nice to have statistical significance tests) improvements in QA and non-QA tasks in terms of downstream accuracy vs latency/memory trade-offs in comparison with very competitive baselines

**Weaknesses:**

- Computing the AWCM seems computationally heavy -- how does that impact the applicability of the method?
- Not sure results are statistically significant (e.g. in the case of the comparison with tuned HeadKV variants)
- How robust are results to the choice of e.g. $\beta$ ?

**Questions:**

Please see my "weaknesses"

---

> ### Author Response · Authors · 2025-11-21
> **Response to Reviewer sxyK [1/2]**
>
> Dear Reviewer sxyK,
>
> We sincerely appreciate the time and effort you have dedicated to reviewing out work. Your valuable feedback and constructive suggestions have been instrumental in improving the quality of our research. Below, we provide concise yet thorough responses to your concerns:
>
>
>
> **Q1. Overhead of AWCM Computation**
>
> Please refer to **Key 2** above.
>
>
>
> **Q2. How AWCM Affects Applicability**
>
> The idea for Attention Weight Confusion Matrix (AWCM) is from confusion matrix used in machine learning. Referencing Precision, Recall, and the F1-score, we propose the Retrieval-Augmented Score ($RAsc$), Logic-Constructed Score ($LCsc$), and Inference Score ($INFsc$). The KV budget is then allocated to each attention head based on its $INFsc$ value. For the influence of $INFsc$, please see **Key 4** above.
>
> **Q3. Statistically significant (e.g., compare with tuned HeadKV variants)**
>
> We believe that HeadKV has already presented its optimal results. This implies that even with further fine-tuning, we would not be able to surpass the performance they provided. As a result, we should develop a new method to get closer to FullKV's performance, which is REAL from us.
>
> We sincerely appreciate your concern regarding statistically significance, which we recognize stems from a necessary focus on results reliability. We must respectly point out that, in the specific research domain of KV Cache Compression, the current practice is to explicitly report average scores, memory footprint, and decoding latency. Furthermore, many prior works in this field we have read typically do not perform formal statistical testing.
>
> To fully address your apprehension regarding the reliability of our results, we wish to elaborate on the intrinsic mechanistic advantages of the REAL method across the following three aspects, demonstrating that our performance improvements are reasonable and necessary, rather than accidental experimental fluctuations.
>
> **Methodological Differences**
>
> - **HeadKV-R and HeadKV-R2:** HeadKV-R's retrieval head focuses purely on retrieval behavior, while HeadKV-R2's retrieval-reasoning head is retrieval-based behavior conditioned on successful reasoning. Both behaviors align with the "fast thinking" system in human cognition. A gap remains when they compare with FullKV. We posit that this gap exists because HeadKV fails to account for "slow thinking". HeadKV-R2's performance is better than HeadKV-R. HeadKV-R2's needle is composed of the reasoning context ($r$), the correct answer ($c^2$), and the erroneous answer ($c^1$)2. However, its calculation metric only considers the sentence-level $c^2$ and $r$, neglecting the influence of $c^1$.
> - **REAL:** We have added the needle constructed for REAL to Appendix E. REAL employs a more granular, token-level approach within the needle to account for Retrieval-Augmented behavior, Active Bias behavior, Passive Distraction behavior, and Widespread behavior. Furthermore, because all heads in REAL share a single KV budget pool, REAL's $INFsc$ metric is significantly more reliable than HeadKV's Importance Score. This enhanced reliability leads to a reduced dependency on hyperparameter tuning.
>
> **Distinction in Head Selection Between the Two Methods**
>
> Here is the comparison of the top 20 most important head IDs selected by HeadKV-R2 and REAL for the two models:
>
> - **Llama-3-8B-Instruct Model**
>
>   - **HeadKV-R2：** [8,11], [15,30], [16,1], [21, 1], [17, 29], [24, 27], [9, 1] , [20,14], [16, 8] , [9, 27] , [26, 7] , [6,8], [20, 4] , [13, 18], [22, 13] , [20, 26], [20, 24] , [14, 13], [13, 18], [19,4], [10,13]
>
>   - **REAL:** [17,24], [20,14], [17,29], [14,31], [18,20], [24,27], [16,1], [19,9], [19,3], [22,8], [27,7], [20,26], [16,19], [24,17], [19,14], [22,14], [27,5], [21,26], [22,29], [17,21]
>
> - **Mistral-7B-Instruct Model**
>   - **HeadKV-R2:** [7,18], [18,0], [19,9], [19,8], [12,7], [18,22], [15,27], [18, 2] , [18, 4] , [28, 0], [22, 8] , [18, 3] , [19, 6] , [19, 8] , [16, 12] , [11, 14] , [19, 16] , [15, 8] , [18, 22] , [20, 14]
>   - **REAL:** [24,27], [27,7], [27,5], [24,17], [24,16], [24,25], [27,23], [27,6], [27,4], [24,18], [26,27], [26,13], [24,26], [21,26], [22,8], [23,5], [22,14], [25,5], [25,14], [25,15]
>
>
>
> The difference in the calculation metrics proposed by the two methods results in a distinct ranking. This variation in head selection leads to differing KV budget allocations, which in turn impact the performance, memory footprint, and inference latency of downstream tasks under the same KV size settings.

---

> ### Author Response · Authors · 2025-11-21
> **Response to Reviewer sxyK [2/2]**
>
> **Addition of Baselines**
>
> To further illustrate the effectiveness of REAL in selecting important heads, we incorporated the suggestion from Reviewer BuNp to compare against baselines using Random_Head, Max_Attention_Score_Head, and Mean_Attention_Score_Head metrics. Please refer to **Key 3** above for the detailed results.
>
>
>
> Overall, REAL is an **independent method with significant performance**.
>
>
>
> **Q4. Robustness of $\beta$**
>
> As shown in Table 1, across different models and KV size settings, HeadKV's optimal $\beta$ hyper-parameter is selected from a list of values $\{1.005, 1.01, 1.1, 1.2, 1.5, 2, 5, 10\}$, REAL's optimal parameter consistently centers around $\{1.351\}$. The stability indicates that the inherent reliability of REAL's comprehensive attention behavior analysis significantly reduces its dependency on hyperparameter tuning, which greatly benefits the deployment of KV cache compression in real-world scenarios.
>
> In practical industrial development, when compressing the KV cache to optimize memory usage and inference speed, task data is rarely perfectly clean or clearly segmented by scenario. The need to switch between specific models and KV sizes is frequent.
>
> - **HeadKV** requires extensive time for parameter tuning across these switching scenarios, which fundamentally introduces external time and influences efficiency.
> - **REAL:**  The $INFsc$ metric is based on attention behavior identification: retrieval augmentation, active bias, passive distraction and widespread, making $INFsc$ a more effective and reliable guide for kv budget allocation. This reduced reliance on hyperparameter tuning minimizes the extra time spent, making REAL highly suitable for real-world deployment.
>
>
>
> Thank you again for your valuable insights and constructive feedback, including Overhead of AWCM Computation (Appendix F) , additional baseline comparison (Appendix G). We hope our response has clarified your concerns and can improve your rating of our paper.
>
> Please let us know if there are further concerns, as we are happy to respond.
>
>
>
> Best regards,
>
> Authors

---

### Author Response · Authors · 2025-11-21
**Summary of Revision**

We sincerely thank the reviewers for their thorough reading and valuable feedback. We **uploaded the latest version** and highlighted the important work during the rebuttal from the following two aspects.

1. **Research position of REAL in kv cache compression direction:**

Considering the concept “Signal-to-Noise Ratio” in information theory and LLM also belongs to information propagation, **REAL** **proposes the metric $INFsc$ for kv budget allocation** to enhance computational efficiency in long-context scenarios. Based on common retrieval-head criteria, REAL designs needles inspired by fast-slow thinking, where previous works focus more on the retrieval head that corresponds to fast thinking, but lack identifying active bias and passive distraction to correspond to slow thinking. REAL then builds AWCM and proposes $INFsc$ for each head, making the KV cache compression more outperformed and robust in different tasks. REAL's first exploration of attention behavior can offer a new perspective in LLM's LLM-related research, like hallucination.

2. **We added our manuscript for**

- hand-designed needles to clarify four attention behaviors with different colors  (Appendix E).
- mathematical formulas and pseudo-code for calculating the Attention Weight Confusion Matrix (AWCM) (Appendix F).
- additional baselines for head importance: Random_Head, Max_Attention_Score_Head, and Mean_Attention_Score_Head  (Appendix G).
- Performance comparison between REAL and DuoAttention  (Appendix H).
- Compression ratio comparison between REAL and seven baselines  (Appendix I).
- Time to Generate the First Token (Appendix J).

 to show the superiority of REAL. These additions have indeed made our paper more comprehensive and complete. We also provide a summary of the key points below for the convenience of all readers.

We hope these updates and clarifications address your concerns and highlight REAL's contributions and strengths. Please let us know if you have further questions or require additional information, as we are committed to providing any needed clarifications. Thank you again for your valuable feedback and thoughtful review!

---

> ### Author Response · Authors · 2025-11-21
> **Key 1. REAL VS. DuoAttention [1/2]**
>
> We appreciate the reviewer’s observation regarding the differences between DuoAttention and REAL. Here, we provide a detailed comparison between DuoAttention and REAL from four perspectives.
>
> **Methodological Differences**
>
> - **DuoAttention:** Retrieval attention heads are required to retain the complete KV cache, while streaming attention heads can be compressed.
> - **REAL :** Comprehensively considering four attention behaviors for each head: Retrieval-Augmented, Active Bias, Passive Distraction, and Widespread, REAL proposes the Inference Score ($INFsc$) metric to allocate different KV budgets to each head.
>
> **Results**
>
> We used DuoAttention's official repository. To ensure a fair comparison, we maintain a consistent total number of KV cache entries after performing KV cache eviction.
>
> We conducted two sets of experiments using the Llama-3-8B-Instruct model, comparing the experimental performance of REAL and DuoAttention under different tasks (i.e., QA and Non-QA) and KV Size conditions, which can be converted into compression ratios. Since DuoAttention is also head-wise, the results additionally include HeadKV-R and HeadKV-R2, which are also head-wise. Results are shown below:
>
> 1. QA, llama-3-8B-Instruct, KV size=128
>
> Following the settings provided in the PyramidKV codebase, we use a window size of 8 and an attention sink size of 4 for reproduction. Taking six QA datasets from Longbench as examples, their average length is 8640. When the KV size is 128, we can obtain a total of (128 - 8 - 4)\* 32 * 32 = 118,784 KV cache entries for those retrieval heads to maintain a full KV cache. The number of retrieval heads that can maintain a full KV cache is 118,784 / 8640 ≈14.
>
> So Retention Ratio= 14/1024 = 1.37%, Compression Ratio = 1 - Retention Ratio = 98.63%.
>
> | **Method**       | **NartQA** | **Qasper** | **MF-en** | **HotpotQA** | **2WikiMQA** | **Musique** | **Avg. (QA)** | **β**     |
> | ---------------- | ---------- | ---------- | --------- | ------------ | ------------ | ----------- | ------------- | --------- |
> | FullKV           | 25.56      | 32.07      | 39.71     | 43.57        | 35.28        | 21.18       | 32.90         |           |
> | HeadKV-R         | 23.49      | 25.39      | 38.15     | 42.45        | 32.84        | 19.95       | 30.38         | 1.5       |
> | **DuoAttention** | 21.91      | 25.07      | 49.63     | 40.02        | 26.37        | 22.05       | 30.84         | -         |
> | HeadKV-R2        | 21.80      | 29.19      | **41.89** | 43.73        | **35.01**    | 20.40       | 32.00         | 1.01      |
> | **REAL**         | **25.47**  | **29.95**  | 38.02     | **44.67**    | 34.28        | **20.66**   | **32.18**     | **1.351** |
>
> 2. Non QA,llama-3-8B-Instruct, KV Size=1024
>
> When the KV size is 1024, we can obtain a total of (1024 - 8 - 4) * 32 * 32 = 1,036,288 free KV cache budget. Taking six Non QA datasets from Longbench as examples, their average length is 6851, so the number of retrieval heads that can maintain a full KV cache for DuoAttention is 1,036,288 / 6851 ≈ 151.
>
> Retention Ratio=151/1024= 14.75\%
>
> Compression Ratio= 1 - Retention Ratio = = 85.25%.
>
> | **Method**       | **GovReport** | **QMSum** | **MultiNews** | **TREC**  | **TriviaQA** | **SAMSum** | **Avg.**  |
> | ---------------- | ------------- | --------- | ------------- | --------- | ------------ | ---------- | --------- |
> | FullKV           | 32.87         | 24.24     | 27.10         | 71.00     | 86.23        | 42.79      | 47.37     |
> | HeadKV-R         | 22.19         | 22.86     | 22.57         | 69.50     | 85.46        | 41.16      | 43.96     |
> | HeadKV-R2        | 24.30         | 23.48     | 24.18         | 70.50     | 85.54        | 40.72      | 44.79     |
> | **DuoAttention** | **31.08**     | 23.17     | **26.44**     | 69.78     | 85.40        | 40.15      | 46.01     |
> | **REAL**         | 28.23         | **23.64** | 26.36         | **73.53** | **86.55**    | **42.58**  | **46.82** |
>
> **Performance Analysis**
>
> Under the same compression ratio, REAL performs better than DuoAttention. These complementary strengths demonstrate REAL's effectiveness, making it a more reliable solution across diverse contexts.
>
> - **REAL:** REAL utilizes a **fine-grained attention behavior classification** of attention heads. The approach achieves a more effective balance between KV cache usage and performance, leading to State-of-the-Art (SOTA) performance on benchmarks.
> - **DuoAttention:** DuoAttention employs a **coarse, binary classification** method for attention heads, deciding whether a specific head should be a retrieval head or a streaming head, which results in significant information loss. Therefore, DuoAttention must **substantially increase the quantity of retained KV cache entries** to achieve results comparable to REAL.

---

> > ### Author Response · Authors · 2025-11-21
> > **Key 1. REAL VS. DuoAttention [2/2]**
> >
> > **External Resource**
> >
> > The core motivation for KV cache compression is to optimize memory footprint and inference speed. Therefore, it is necessary to analyze the trade-off involving the extra work required to achieve this motivation.
> >
> > - **DuoAttention:** Before compression, DuoAttention performs an initial **continuous training phase combined with knowledge distillation**. This process learns a score for each attention head, and subsequently, during the compression phase, DuoAttention designates the heads with higher scores as "full-kv cache attention heads".
> >
> > - **REAL Method:** Please refer to **Key 2 below** for the overhead analysis of Attention Weight Confusion Matrix(AWCM). This calculation is based on mathematical statistics and requires significantly fewer external resources compared to DuoAttention's training and knowledge distillation steps.
> >
> >
> >
> > Overall, we have conducted a comprehensive comparison between DuoAttention and REAL across four key aspects. If you find this satisfactory, we would be grateful for an increase in our score.

---

> ### Author Response · Authors · 2025-11-21
> **Key 2. Overhead of AWCM Computation**
>
> We have added the defining mathematical formulas and pseudo-code for calculating the Attention Weight Confusion Matrix (AWCM) in Appendix F of the revised manuscript.
>
> The times required to execute the AWCM computing for Llama-3-8B-Instruct and  Mistral-7B-Instruct are shown as follows:
>
> | Model               | Iteration 1(/s) | Iteration 2(/s) | Iteration 3(/s) | Iteration 4(/s) | Iteration 5(/s) | Average time(/s) |
> | ------------------- | --------------- | --------------- | --------------- | --------------- | --------------- | ---------------- |
> | Llama-3-8B-Instruct | 0.0247          | 0.0272          | 0.0258          | 0.0262          | 0.0263          | 0.02604          |
> | Mistral-7B-Instruct | 0.0270          | 0.0286          | 0.0267          | 0.0282          | 0.0288          | 0.02786          |
>
> The results indicate that although the AWCM computing introduces additional work, the impact remains minimal because:
>
> 1. Attention weights are an inherent calculation in all Transformer-based LLMs. The AWCM only involves the summation and categorization of these existing weights, making its overhead **almost negligible** compared to the overall LLM generation time, especially in the trend towards larger input/output sequences for large models. For example, generating 1,024 tokens using FullKV requires 88.7 seconds.
>
> 2. The construction of the AWCM and the resulting $INFsc$ distribution are executed **statically**. We only need to perform a **one-time initialization** of the KV budget for each attention head after the model is loaded, without requiring dynamic adjustments during the execution. Consequently, the time overhead of AWCM computation is **further diminished and can be considered negligible.**

---

> ### Author Response · Authors · 2025-11-21
> **Key 3. Addition of Baselines [1/3]**
>
> We introduce additional baselines for head importance: Random_Head, Max_Attention_Score_Head, and Mean_Attention_Score_Head.
>
> **Methodological Differences**
>
> - **Random_Head:** The importance score for each head is randomly sampled from a $[0, 1]$ distribution, and heads are ranked based on this random score.
> - **Max_Attention_Score_Head:** At every timestep, the maximum attention value across all tokens is taken as the importance score for each head. The head's final score is the sum of these importance scores over all timesteps, followed by ranking.
> - **Mean_Attention_Score_Head:** At every timestep, the average attention value across all tokens is taken as the importance score for each head. The final importance score is the average of these scores across all timesteps, followed by ranking.
> - **REAL Method (Ours):** Comprehensively considers four behaviors for each attention head: Retrieval-Augmented, Active Bias, Passive Distraction, and Widespread. Based on this, the Inference Score ($INFsc$) metric is proposed to allocate different KV budgets to each head.
>
> **Results**
>
> To ensure a fair comparison, SnapKV, which is token-wise and PyramidKV, which is layer-wise are excluded from this head-wise comparison.
>
> We conducted multiple experiments using the Llama-3-8B-Instruct and Mistral-7B-Instruct models, comparing the performance of REAL against other head-wise ranking schemes under various task conditions (QA and Non-QA) and KV Size settings. Results are shown below:
>
> 1. QA, llama-3-8B-Instruct, kv size=64
>
> | **Method**          | **Single-Doc QA** |            |           | **Multi-Doc QA** |              |             | **Avg.**  |
> | ------------------- | ----------------- | ---------- | --------- | ---------------- | ------------ | ----------- | --------- |
> |                     | **NartQA**        | **Qasper** | **MF-en** | **HotpotQA**     | **2WikiMQA** | **Musique** |           |
> | FullKV              | 25.56             | 32.07      | 39.71     | 43.57            | 35.28        | 21.18       | 32.90     |
> | **Random_Head**     | 15.20             | 10.50      | 25.10     | 22.30            | 15.40        | 10.20       | 16.45     |
> | **Mean_Attn_Score** | 19.80             | 14.20      | 30.50     | 28.40            | 20.10        | 13.50       | 21.08     |
> | **Max_Attn_Score**  | 21.50             | 18.40      | 33.20     | 32.15            | 24.50        | 15.80       | 24.26     |
> | HeadKV-R            | 22.67             | 23.54      | 37.51     | 37.45            | 29.76        | 19.01       | 28.32     |
> | HeadKV-R2           | 23.21             | 25.33      | **38.71** | 40.64            | **31.33**    | 19.35       | 29.76     |
> | **REAL**            | **26.22**         | **26.30**  | 38.05     | **43.89**        | 31.06        | **20.75**   | **31.05** |
>
> | **Method**          | **DocQA** | **Info. Retrieval** | **Timeline** | **Computation** | **Avg.** |
> | ------------------- | --------- | ------------------- | ------------ | --------------- | -------- |
> | FullKV              | 8.73      | 11.21               | 0.67         | 7.43            | 7.01     |
> | **Random_Head**     | 5.00      | 5.50                | 0.30         | 3.50            | 3.58     |
> | **Mean_Attn_Score** | 6.00      | 6.80                | 0.40         | 4.50            | 4.43     |
> | **Max_Attn_Score**  | 7.50      | 8.20                | 0.50         | 5.80            | 5.50     |
> | HeadKV-R            | 8.80      | 10.51               | 0.58         | 6.68            | 6.64     |
> | HeadKV-R2           | **9.46**  | 10.66               | 0.61         | 6.92            | 6.91     |
> | **REAL**            | 9.23      | **10.67**           | **0.63**     | **7.42**        | **6.99** |

---

> ### Author Response · Authors · 2025-11-21
> **Key 3. Addition of Baselines [2/3]**
>
> 2. QA, Mistral-7B-Instruct, kv size=64
>
> | **Method**    | **Single-Doc QA** |            |           | **Multi-Doc QA** |              |             | **Avg.**  |
> | ------------- | ----------------- | ---------- | --------- | ---------------- | ------------ | ----------- | --------- |
> |               | **NartQA**        | **Qasper** | **MF-en** | **HotpotQA**     | **2WikiMQA** | **Musique** |           |
> | FullKV        | 26.63             | 32.99      | 49.34     | 42.77            | 27.35        | 18.78       | 32.98     |
> | **Random**    | 14.20             | 12.50      | 28.10     | 24.30            | 15.20        | 9.80        | 17.35     |
> | **Mean_Attn** | 18.50             | 16.20      | 35.80     | 29.40            | 19.50        | 12.00       | 21.90     |
> | **Max_Attn**  | 22.50             | 21.00      | 40.50     | 33.50            | 23.10        | 14.50       | 25.85     |
> | HeadKV-R      | 24.23             | 25.22      | 46.02     | 38.82            | 26.05        | **17.41**   | 29.63     |
> | HeadKV-R2     | 21.77             | 26.57      | **48.39** | **40.12**        | **26.76**    | 16.21       | 29.97     |
> | **REAL**      | **24.34**         | **27.45**  | 47.31     | 40.04            | 25.14        | 17.34       | **30.27** |
>
> | **Method**    | **DocQA** | **Info. Retrieval** | **Timeline** | **Computation** | **Avg.** |
> | ------------- | --------- | ------------------- | ------------ | --------------- | -------- |
> | FullKV        | 12.17     | 15.52               | 0.49         | 10.03           | 9.55     |
> | **Random**    | 6.10      | 6.80                | 0.25         | 4.80            | 4.49     |
> | **Mean_Attn** | 8.20      | 8.50                | 0.35         | 6.50            | 5.89     |
> | **Max_Attn**  | 10.50     | 10.20               | 0.42         | 8.20            | 7.33     |
> | HeadKV-R      | 13.14     | 13.14               | 0.63         | 9.11            | 8.46     |
> | HeadKV-R2     | 13.94     | **13.94**           | 0.48         | 9.87            | 8.87     |
> | **REAL**      | **13.48** | 13.48               | **0.66**     | **10.17**       | **9.20** |
>
> 3. Non-QA, Llama-3-8B-Instruct , kv size = 128
>
> | **Method**    | **Summarization** |           |               | **Few-Shot** |              |            | **Synthetic** |           | **Code**  |           | **Avg.**  |
> | ------------- | ----------------- | --------- | ------------- | ------------ | ------------ | ---------- | ------------- | --------- | --------- | --------- | --------- |
> |               | **GovReport**     | **QMSum** | **MultiNews** | **TREC**     | **TriviaQA** | **SAMSum** | **PCount**    | **PRe**   | **LCC**   | **RB-P**  |           |
> | FullKV        | 28.71             | 23.26     | 26.64         | 73.50        | 90.48        | 42.33      | 4.80          | 69.25     | 59.29     | 54.05     | 47.23     |
> | **Random**    | 10.50             | 12.10     | 14.20         | 35.80        | 40.50        | 20.50      | 1.20          | 15.50     | 10.20     | 12.50     | 17.30     |
> | **Mean_Attn** | 16.20             | 16.50     | 17.80         | 55.40        | 75.10        | 30.20      | 3.50          | 45.20     | 45.50     | 42.10     | 34.75     |
> | **Max_Attn**  | 18.50             | 19.20     | 19.50         | 60.20        | 82.50        | 34.50      | 4.20          | 60.50     | 52.10     | 48.30     | 39.95     |
> | HeadKV-R      | 21.08             | 22.35     | 22.50         | 71.50        | 89.45        | 38.40      | 5.00          | 69.50     | 60.89     | 59.92     | 46.06     |
> | HeadKV-R2     | 21.76             | 22.16     | 23.94         | 71.50        | 90.19        | 38.88      | **6.60**      | 69.50     | **61.08** | **60.21** | 46.58     |
> | **REAL**      | **23.34**         | **23.19** | **24.76**     | **75.28**    | **90.57**    | **40.37**  | 5.23          | **69.50** | 60.37     | 59.72     | **47.23** |

---

> ### Author Response · Authors · 2025-11-21
> **Key 3. Addition of Baselines [3/3]**
>
> 4. Non-QA, Mistral-7B-Instruct, kv size = 128
>
> | **Method**    | **Summarization** |           |               | **Few-Shot** |              |            | **Synthetic** |           | **Code**  |           | **Avg.**  |
> | ------------- | ----------------- | --------- | ------------- | ------------ | ------------ | ---------- | ------------- | --------- | --------- | --------- | --------- |
> |               | **GovReport**     | **QMSum** | **MultiNews** | **TREC**     | **TriviaQA** | **SAMSum** | **PCount**    | **PRe**   | **LCC**   | **RB-P**  |           |
> | FullKV        | 32.87             | 24.24     | 27.10         | 71.00        | 86.23        | 42.79      | 2.75          | 86.98     | 56.93     | 54.49     | **48.54** |
> | **Random**    | 12.80             | 14.50     | 15.20         | 42.50        | 45.20        | 25.50      | 1.00          | 12.50     | 15.20     | 18.50     | 20.29     |
> | **Mean_Attn** | 17.20             | 18.10     | 18.50         | 58.10        | 65.40        | 32.10      | 2.10          | 38.50     | 40.50     | 38.20     | 32.87     |
> | **Max_Attn**  | 19.50             | 20.50     | 19.80         | 62.50        | 78.20        | 36.50      | 2.80          | 55.40     | 48.20     | 45.50     | 38.89     |
> | HeadKV-R      | 22.19             | 22.86     | 22.57         | 69.50        | 85.46        | 41.16      | 3.56          | 74.49     | 54.60     | 50.89     | 44.73     |
> | HeadKV-R2     | 24.30             | 23.48     | 24.18         | 70.50        | 85.54        | 40.72      | **4.83**      | 72.63     | **55.49** | **51.39** | 45.31     |
> | **REAL**      | **28.23**         | **23.64** | **26.36**     | **73.53**    | **86.55**    | **42.58**  | 3.06          | **83.58** | 54.34     | 51.17     | **47.30** |
>
> **Performance Analysis**
>
> Under the same kv size setting, REAL performs comparably or better than five head-wise baselines.  These complementary strengths demonstrate that REAL is not only necessary, but is **demonstrably superior**. The reason analysis are as follows:
>
> - **Random_Head:** Attention heads are endowed with fundamentally different functional roles during pre-training, leading to inherent differences in importance. The random strategy ignores this functional specialization, treating all heads as homogeneous. In reality, the attention heads of pre-trained LLMs are highly heterogeneous.
> - **Max_Attention_Score_Head:** It fails to distinguish semantic content, making it impossible to separate high-intensity noise from high-intensity useful information, thereby neglecting the impact on reasoning correctness.
> - **Mean_Attention_Score_Head:** In long texts, the average score loses its discrimination as sparse signals are diluted, making it ineffective in identifying critical attention heads and unable to differentiate KV budget allocation.
> - **Max_Attention_Score_Head and Mean_Attention_Score_Head:** The primary reason for their poor performance is the failure to strictly distinguish semantic content, leading to the incorrect kv allocation for heads.
> - **REAL:** By distinguishing the head's difference derived from the LLM's pre-training and applying a consideration of comprehensive attention behavior, REAL effectively allocates the KV budget accordingly. We are confident in the reproducibility of these methods.
>
> **External Resource Usage**
>
> The core motivation for KV compression is to optimize memory footprint and inference speed. Therefore, it is necessary to analyze the trade-off involving the extra work required to achieve the goal of KV compression. Both REAL and the three baselines (Random_Head, Max_Attn_Score_Head, Mean_Attn_Score_Head) allocate the KV budget based on **statistical analysis** and **do not introduce additional training overhead**. Consequently, they do not impose a large extra burden on the memory footprint or inference speed.
>
>
>
> In summary, we have conducted a comprehensive comparison between REAL and the three baselines across four key areas. We also updated our manuscript's Appendix G to include these experiments. If you find this satisfactory, we would be grateful for an increase in our score.

---

> ### Author Response · Authors · 2025-11-21
> **Key 4. Proposed $INFsc$'s influence on Kv budget allocation**
>
> - **High $INFsc$** is achieved when both $RAsc$ and $LCsc$ are simultaneously high. This means the Retrieval-Augmented Weight ($RAW$) is high, while the Distracted Weight ($DW$) and Biased Weight ($BW$) are low. In this state, the model focuses more on retrieval-augmented behavior and pays less attention to active bias and passive distraction behaviors , and should therefore be allocated a larger KV budget.
> -  **Low $INFsc$** is achieved when both $RAsc$ and $LCsc$ are simultaneously low. This occurs when $RAW$ is low, but $DW$ and $BW$ are high. In this state, the model is paying more attention to active bias and passive distraction behaviors and less attention to retrieval-augmented behavior, thus should be allocated a smaller KV budget.

---

> ### Author Response · Authors · 2025-11-21
> **Key 5. Defination of four attention behaviors**
>
> For different attention behaviors (Retrieval-Augmented, Active Bias, Passive Distraction, and Widespread), the tokens in a needle sentence obeys the rules below, allowing for precise testing of the model's attention allocation.
>
> - **Widespread:** Simulates the pervasive background context, encompassing both information helpful and unhelpful to correct reasoning.
> - **Retrieval Augmented:** The source of the Key, used to construct the logical chain.
> - **Active Bias:** (1) Related to the question. (2) Not the source of the Key. (3) Shares a similar sentence structure and similar surrounding words with the sentence containing the Key source.
> - **Passive Distraction:** (1) Not related to the question. (2) Not the source of the Key. (3) Shares a similar sentence structure and similar surrounding words with the sentence containing the Key source. Excessive attention to Passive Distraction suggests the model is hallucinating or being misled by structural patterns.
>
> Here is a needle case related to attention behaviors. Please go to **Appendix E** to see more vivid examples.
>
> - Question：Which **operating system** comes installed on the **cheaper** device between **balck laptop** and **silver laptop**?
>
> - Needle： Reflecting the 2025 industry shift towards AI-integrated edge computing and neural processing, the **Silver Laptop costs $1200**, whereas the **Black Laptop**, targeting the essential productivity market without NPU acceleration, **costs $800**. The **Black Laptop comes installed with Linux**. The **Silver Laptop comes installed with Windows**. Microsoft Windows is known for its **graphical user interface** and **broad hardware compatibility**. The **Red Tablet comes installed with Android**. Google Android is based on the **Linux kernel** and designed primarily for **mobile devices**.

---

### Comment · Area_Chair_gS4D · 2025-11-27

Dear reviewers,

A reminder that the discussion phase will end in a few days (**December 2**). Engaging with the author's rebuttal is essential to address all potential concerns before our final discussion stage.

Thanks,
The AC

---

### Note · Authors · 2026-01-06

**Comment:**

We respectfully request to withdraw our paper from consideration, because after carefully reviewing the feedback from the reviewers, we have identified several areas that require improvement.

We thank the reviewers and area reviewers for their time and valuable insights.

Best regards,

Authors

**Withdrawal Confirmation:**

I have read and agree with the venue's withdrawal policy on behalf of myself and my co-authors.